# A multimodal cross-species comparison of pancreas development

Kaiyuan Yang [1,2,18], Hannah Spitzer [3,4,18], Michael Sterr[1,2], Karin Hrovatin [3,5], Sean de la O[6,7,8], Xinghao Zhang[9,10], Eunike Sawitning Ayu Setyono [1,2], Minhaz Ud-Dean[11], Thomas Walzthoeni[11], Krzysztof Flisikowski[12], Tatiana Flisikowska [12], Angelika Schnieke [12], Katharina Scheibner[1,2], James M. Wells [9,10,13], Julie B. Sneddon [6,7,8], Barbara Kessler[2,14,15], Eckhard Wolf [2,14,15], Elisabeth Kemter [2,14,15], Fabian J. Theis [3,5,16] ✉ & Heiko Lickert [1,2,17] ✉

Human pancreas development remains incompletely characterized due to restricted sample access. We investigate whether pigs resemble humans in pancreas development, offering a complementary large-animal model. As pig pancreas organogenesis is unexplored, we first annotate developmental hallmarks throughout its 114-day gestation. Building on this, we construct a pig single-cell multiome pancreas atlas across all trimesters. Cross-species comparisons reveal pigs resemble humans more closely than mice in developmental tempo, epigenetic and transcriptional regulation, and gene regulatory networks. This further extends to progenitor dynamics and endocrine fate acquisition. Transcription factors regulated by NEUROG3, the endocrine master regulator, are over 50% conserved between pig and human, many being validated in human stem cell models. Notably, we uncover that during embryonic development, emerging beta-cell heterogeneity coincides with a species-conserved primed endocrine cell (PEC) population alongside NEUROG3-expressing cells. Overall, our work lays the foundation for comparative investigations and offers unprecedented insights into evolutionarily conserved pancreas organogenesis mechanisms across animal models.

Mice have been the main mammalian model to investigate the evolutionarily conserved basic principles of development and disease. As human fetal samples are largely inaccessible due to ethical and practical restrictions, our knowledge of pancreas development is mainly built on extensive investigations in mice[1,2]. In brief, from foregut endoderm, ventral and dorsal pancreatic buds are specified, evaginated, and fused over the midline to form the organ anlage. This multipotent progenitor epithelium undergoes apical-basal polarization, microlumen formation, single-layer stratification, and branching morphogenesis, which is concomitant with tip-trunk patterning and differentiation to form the acinar, ductal, and endocrine compartments of the final pancreas (Fig. 1a)[1,3–7]. With this blueprint, the signals and factors orchestrating in vivo development are applied to human pluripotent stem cell differentiation in vitro to generate endocrine islet cells, such as glucagon-producing alpha cells and insulin-producing beta cells, enabling stem cell-derived islet replacement therapy to treat diabetes[7–10]. However, these protocols cannot produce fully functional islet cells while eliminating undifferentiated and off-target cell types[11,12], partly resulting from the inherent differences in developmental timescales, neighboring tissue interactions, and spatiotemporal gene expression and regulation between mouse and human pancreas differentiation and morphogenesis[13]. These differences highlight the need for cross-species comparisons with additional model systems to uncover

conserved mechanisms of organogenesis and to bridge the translational gap between mice and humans.

Pigs have coevolved with humans over the past 10,000 years, a period when pig domestication coincided with human agricultural civilization[14]. Despite diverging from humans earlier than mice (94 vs. 87 million years ago)[15], pigs retain genomic feature similarity to humans compared to the rapidly evolving mouse lineage[16]. Moreover, as omnivorous animals, pigs resemble humans in metabolism and physiology. Pig organs share anatomical and functional features with humans, making them a favored option for xenotransplantation with compatible organ size and fewer ethical concerns compared to non-human primates[17]. Porcine islets show transcriptional characteristics similar to human islets[18] and represent a potential source for xenotransplantation[19], since pigs and humans have similar glycemic control and identical insulin amino acid sequence. This allowed insulin therapy using pig insulin before recombinant human insulin became available[20–22]. However, whether these shared traits position pigs as a relevant model for human pancreas development remains unexplored, particularly regarding the molecular and (epi)genetic mechanisms driving embryonic development and organogenesis. Given pigs' anatomical and physiological similarities to humans and their extended gestational period (114 days vs. 21 in mice, 280 in humans), we reasoned that pigs could serve as a large animal model to complement the existing rodent models, bridging the translational gap in understanding pancreas development from mice to humans.

Here, we utilize temporally resolved single-cell multi-omics to compare pancreas development in mice, pigs, and humans. Our analysis demonstrates the complementary potential of the pig model for identifying both species-specific and evolutionarily conserved mechanisms of pancreas organogenesis, morphogenesis, and differentiation.

## Results
### The developing pig pancreas
To compare pancreas development across species (Fig. 1a), we first defined the unexplored hallmarks of pig pancreas development during the 114-day gestation by identifying major lineage markers known in mouse and human (Fig. 1b–g). The pig pancreas primordium emerged around embryonic day (E)18, when the earliest pancreatic transcription factor (TF) PDX1 was detected in both ventral and dorsal foregut, with the former having a thicker evaginating layer of cells as reported in human embryos (Supplementary Fig. 1a)[23]. At E20, pancreas organogenesis initiated with the apparent dorsal and ventral buds (T1, Fig. 1a) that comprised an unpolarized multi-layered progenitor epithelium, harboring glucagon+ alpha cells and very few insulin+ beta cells (Fig. 1b and Supplementary Fig. 1b). The early emergence of hormone+ cell resembles the first wave of Neurogenin-3 (NEUROG3)-mediated endocrinogenesis in mice during the primary (1°) transition[5], which is absent at the corresponding stage in human when the endocrine transcription factors NEUROG3 and NKX2-2 remain undetectable[23]. Upon the fusion of the two pancreatic buds at E30, the expression of NEUROG3 protein and transcripts diminished from the pancreas (Fig. 1c and Supplementary Fig. 1c, h). At E40, concurrent with pancreatic epithelial polarization, stratification and tip-trunk patterning (T2, Fig. 1a), NEUROG3 reappeared, initiating the second wave of endocrinogenesis comparable to the secondary (2°) transition in mice (Fig. 1d and Supplementary Fig. 1d). By E54, CPA1+ tip and SOX9+ trunk domains were clearly established (Fig. 1e and Supplementary Fig. 1e). SOX9 remained faintly detectable in CPA1+ tip cells that were predisposed to acinar fate at E63, while endocrine cell clusters started to appear (Fig. 1f and Supplementary Fig. 1f). From E85 onwards, endocrine (NKX6-1+), ductal (SOX9+) and acinar (GATA4+) compartments continued to segregate, with these TFs showing almost exclusive expression patterns (Fig. 1g and Supplementary Fig. 1g, j). During this time, pig proto-islets appeared (T3, Fig. 1a) and formed an

intermingled islet architecture near birth (Supplementary Fig. 1k), which resembles the postnatal islet architecture in human, but not the typical core (beta cells)-mantle (alpha cells) islet structure known from mice[24–26].

Aligning the timing of pancreas development milestones (i.e. time points labeled in Fig. 1a) in human, pig and mouse allowed us to compare the developmental tempo across species (Fig. 1h). Overall, pig pancreas morphogenesis and differentiation speed showed a closer resemblance to humans when compared to mice, particularly during the 2° transition. The formation of the pancreatic anlage in the form of two buds (T1) occupied 10% of the duration of human gestation, 12% in mouse and 17% in pig. In contrast, pancreatic morphogenesis (T2) and islet formation (T3) progressed much faster in mice (42%), contrary to the longer duration in human (82%) and pig (65%), when both species underwent extended acinar terminal differentiation and islet remodeling[23].

To capture the dynamic transcriptional changes of lineage allocation during pig pancreas development, we performed 10X single-cell RNA sequencing (scRNA-seq) on 124,869 cells isolated from pancreata across all three trimesters (Supplementary Fig. 2a and Supplementary Data 1). The resulting dataset was subjected to stringent doublet removal: cells consistently identified as doublets by >3 among the six methods used were removed, and entire clusters with a doublet frequency >70% were excluded (see Methods for details). From this filtered dataset, we extracted pancreatic epithelial cells that co-expressed CDH1 and EPCAM (Supplementary Fig. 2b) and identified eight clusters (Fig. 1i, j, Supplementary Fig. 2c, d and Supplementary Data 2): Ductal (SOX9/SLC4A4), Acinar (CPA1/CEL), NGN3 (NEUROG3/TUBB2B, endocrine progenitors, NGN3 is used as a cluster name to differentiate from NEUROG3 protein and/or mRNA), FEV (FEV/CHGB, endocrine precursors with low NEUROG3 expression), Beta (INS/G6PC2), Alpha (GCG/IRX1), Delta (SST/HHEX), and PP (PPY/ETV1) cell clusters. Among the top ranked genes, published human markers were also found in the pig pancreas, such as MDK in the NGN3, DDC in FEV, ASB9 in Beta, TTR in Alpha, RBP4 in Delta cell clusters[27–30]. Diabetes risk related genes were expressed in a cell-type specific manner, e.g. ABLIM1 in Ductal, TAGLN3 in NGN3, DIRAS3 in Alpha, CFAP61 in all endocrine cell clusters[31–34]. We further identified two molecularly distinct cell clusters: 1) A predicted multipotent progenitor cell (MPC) cluster, which transiently existed during E23-40 and co-expressed key pancreatic progenitor TFs (PDX1, PTF1A, SOX9, NKX6-1, PROX1). This predicted MPC cluster resolved from E54 onwards, when exocrine and endocrine lineages clearly separated and committed to differentiation; 2) A primed endocrine cell (PEC) cluster first emerged at E23 alongside sparse NEUROG3+ endocrine progenitors. This population, which persisted throughout all subsequent stages, exhibited features of endocrine cells and expressed genes coding for cytoskeletal components (TUBA1B, TUBB, STMN1) and cell cycle regulators (H2AFZ, PCLAF).

The emergence of the identified cell clusters in the scRNA-seq atlas was consistent with our immunofluorescence analysis of the pig pancreas, thereby verifying the cell type annotations. For instance, the NGN3 cluster contained only two cells in the scRNA-seq atlas at E33, when the NEUROG3 protein and transcript were almost undetectable in the pancreas (Fig. 1c, i and Supplementary Fig. 1c, h); whereas the CPA1+ Acinar, SST+ Delta and PPY+ PP clusters appeared after E40, when these proteins could be detected in the pancreas (Fig. 1i and Supplementary Fig. 1d, e, i).

### Cross-species multiome atlas comparison
To compare the transcriptional features of all epithelial pancreatic cell types across species, we generated atlases of human and mouse pancreas development by integrating published 10X scRNA-seq datasets[30,35–42]. The human atlas had 188,488 cells from 7-20 week-post-conception (wpc), while the mouse atlas had 135,575 cells covering E8.5-18.5. Cells were clustered, annotated, and the pancreatic

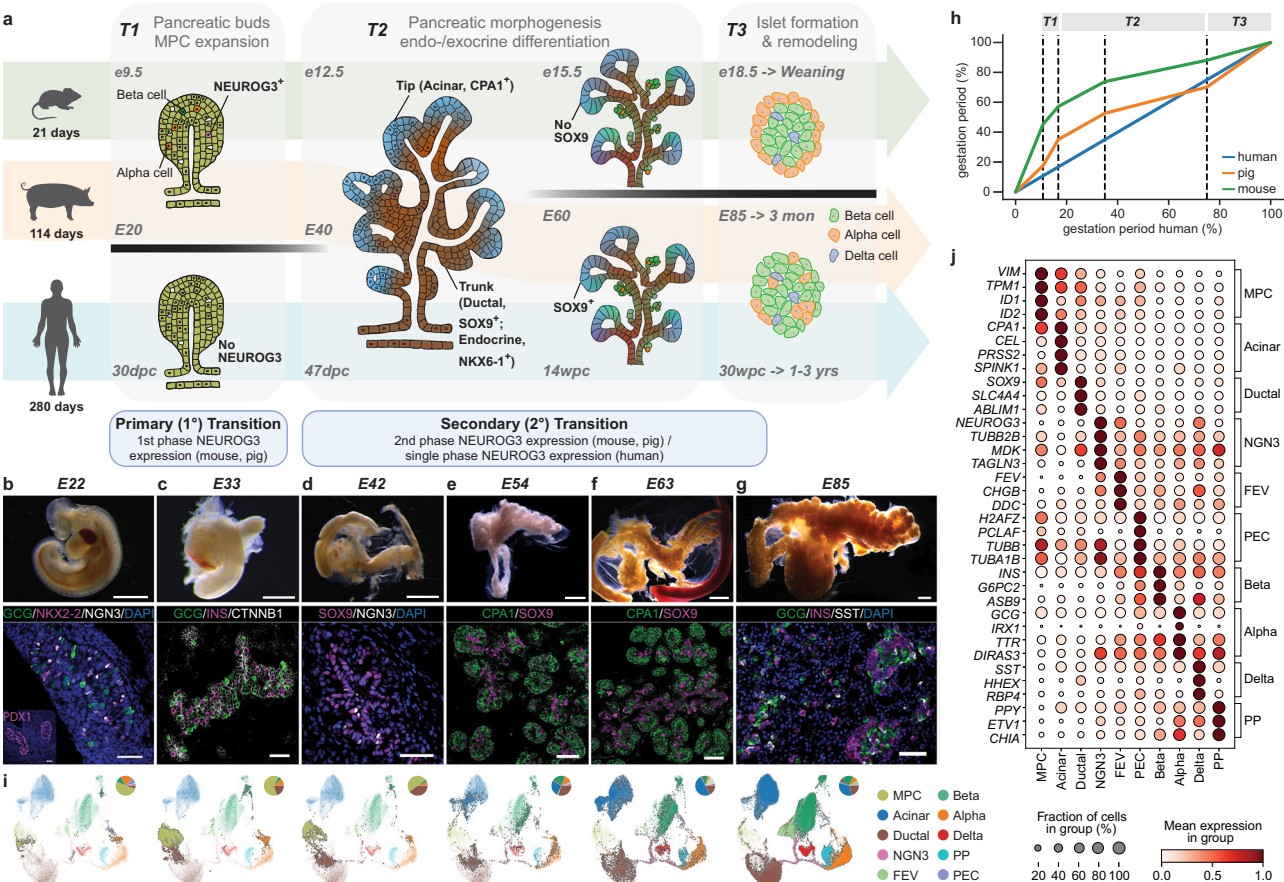

**Fig. 1 | A detailed roadmap of pig pancreas organogenesis. a** Schematic comparison of pancreas organogenesis in mouse, pig and human[1,6,7]. Pancreas organogenesis initiates when the pancreatic buds (dorsal bud shown) emerge from the foregut endoderm while predicted multipotent pancreatic progenitor cells (MPC) expand into a multi-layer epithelium (T1). In mouse and pig but not human, neurogenin 3-induced endocrinogenesis marks the primary (1°) transition, generating alpha cells and few beta cells. Pancreatic morphogenesis (T2) occurs after the fusion of the pancreatic buds. The epithelium undergoes polarization, microlumen formation and coalescence into the near single-layer epithelial tree, which is subsequently patterned into trunk and tip domains during branching morphogenesis. This coincides with progenitor differentiation forming exocrine and endocrine compartments, referred to as secondary (2°) transition when the bulk of beta cells emerge. During T3, the pancreatic ductal and acinar cells proliferate and terminally differentiate, while delaminating endocrine cells form proto-islets of various sizes.

**b**–**g** Bright-field images of wild-type pig embryos or embryonic pancreas (scale bar 2 mm) and immunofluorescence identification of lineage markers highlighting differentiation events in tissue sections (scale bar 50 μm) at different time points. Images are representative of 3 samples per time point. **h** Schematic comparison of mouse and pig development speeds relative to human, showing the timing of each pancreas developmental milestone (labeled in **a**) as a percentage of gestation duration marked with dashed lines. **i** Uniform Manifold Approximation Projection (UMAP) plots showing integrated pig pancreas atlas and cluster changes from E22-85. Cells at each developmental stage (see **b**-**g**) are highlighted and colored by cell type, with a pie chart of relative cell type composition at the upper right corner. **j** Dot plot showing mean gene expression of marker genes for each cluster of the integrated pig pancreas atlas. **i** and **j**: scRNA-seq of pancreatic cells from wild-type and *INS*-eGFP pigs. Detailed sample information is provided in Supplementary Data 1.

epithelial lineage selected as for the pig dataset (Supplementary Fig. 3a–f). All pig pancreatic cell clusters were identified in human and mouse datasets except the PEC cluster, which was absent potentially due to its low abundance in the analyzed samples (Fig. 2a, Supplementary Fig. 3g, h and Supplementary Data 2). PP (*PPY/ETV1*) cells did not separate from the Alpha cluster in mouse (Supplementary Fig. 4a). *FEV*+ cells in human were annotated together with the NGN3 cluster as EP (endocrine progenitor) cluster due to high *NEUROG3* expression (Supplementary Fig. 4b). The Epsilon (*GRHL*) cluster was identified in human and mouse datasets, but surprisingly not in the pig pancreas. The Epsilon marker genes from human and mouse did not show significant enrichment in any cell clusters in the pig (Supplementary Fig. 4c).

To assess the comparability of pancreatic cell types across species, we performed a correlation analysis using the average expression of species-conserved highly variable genes in each cluster, which reflected the global transcriptional programs of each species. Pig and human clusters had an overall stronger correlation (r = 0.6-0.7),

whereas MPC, Beta, and EP clusters in mouse showed the lowest correlation to human clusters (r = 0.45-0.54; Fig. 2b and Supplementary Fig. 4d). We further examined the significantly upregulated differentially expressed genes (DEGs) of each cluster across species, which can capture the subtle species-specific differences in gene expression. Although the pig genome annotation is not as complete as that of human and mouse, the MPC, Beta, and Ductal clusters of pig and human had more overlapping total DEGs and TFs. In contrast, in the Acinar and Alpha clusters, mouse and human shared more total DEGs, but not TFs (Supplementary Fig. 4e). Each species displayed distinct cell-type-specific expression patterns of DEGs (Fig. 2c, Supplementary Fig. 4f, and Supplementary Data 3). For example, the top TFs *SOX6* and *SOX9* of human MPC cluster showed enriched expression in the Ductal cluster of pig and mouse. The expression patterns of the top TFs in endocrine progenitors were relatively conserved among the three species. Among the top TFs of the human Beta cluster, the maturation TF *MAFA* was detected in pig but absent in mouse beta cells; *PLAGL1*, related to transient neonatal diabetes[43,44], appeared in pig beta cells,

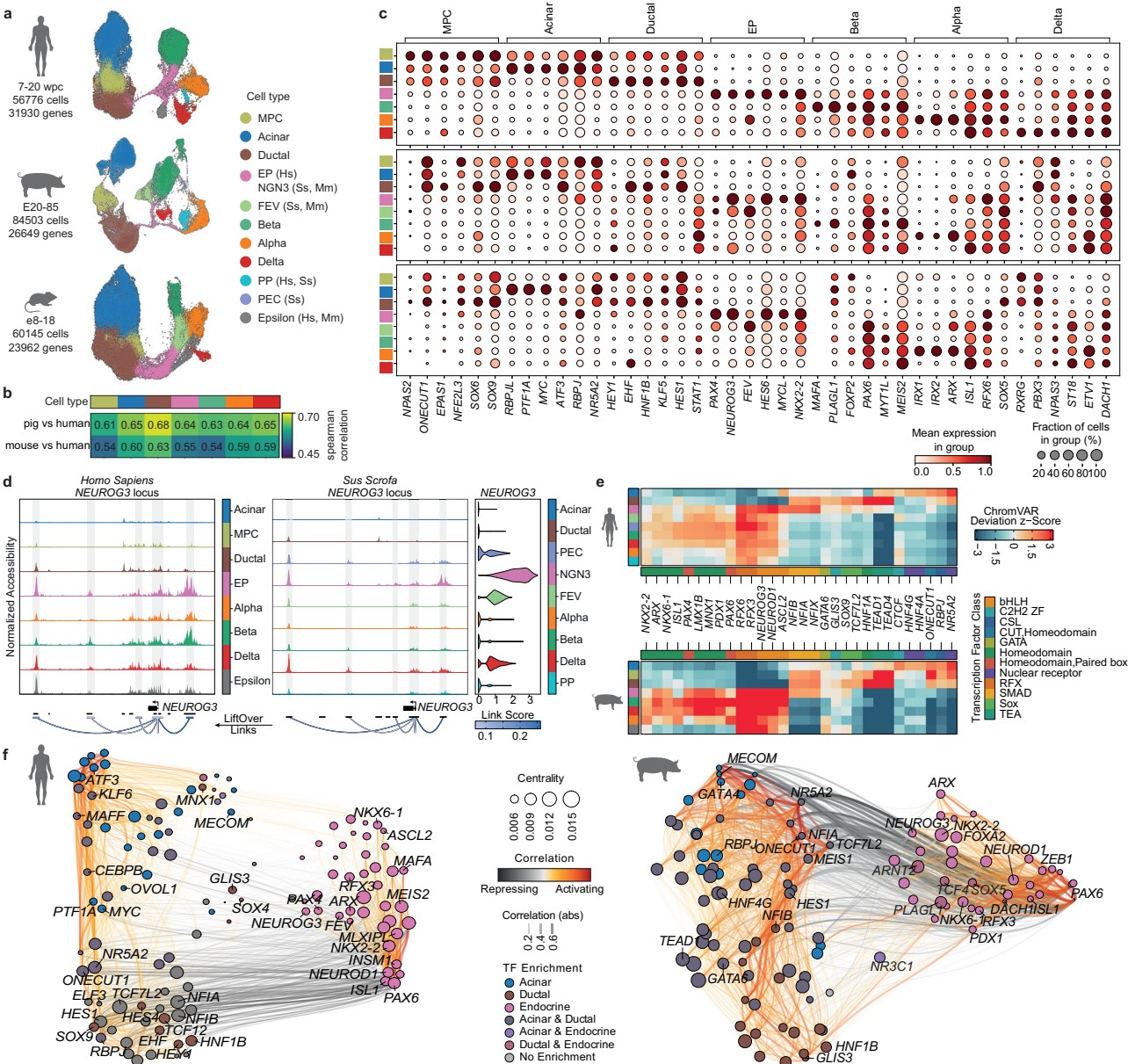

**Fig. 2 | Cross-species multiome atlas comparison reveals conservation between species. a** UMAPs of integrated human, pig, and mouse atlases of pancreas development. Cells were colored by cell type, with the same colors indicating the same cell types across species (Hs, *Homo sapiens*; Ss, *Sus scrofa*; Mm, *Mus musculus*). **b** Spearman correlation of mean normalized gene counts per cluster comparing pig-human and mouse-human pairs (based on human orthologs). Analysis is limited to the 851 genes shared in the intersection of all species' 4000 highly variable genes. **c** Dot plot showing mean expression of cell-type-specific differentially expressed TFs in human (one-vs-rest analysis using edgeR; FDR-corrected p-value < 0.05; Supplementary Data 3). Top genes with the highest logfold change that are expressed in > 20% of cells of the cluster are shown across all clusters (square color = cell type color in **a**) for all three species. Genes are mapped to human orthologs. Dot size represents the fraction of expressing cells per cluster (logarithmic scale). **d** Coverage plots showing pig-human conserved *NEUROG3* genomic regions. Link positions were converted from pig to human genome assembly with UCSC liftOver tool (http://genome.ucsc.edu)[119]. Pseudo-bulk accessibility tracks were used to visualize DNA accessibility in a region by averaging signals from all cells within a cluster. **e** Heatmap of top differentially active motifs across cell types computed with chromVAR[121] using human 12wpc scATAC-seq (top) and pig multiome (bottom) datasets. **f** UMAP visualizations of the inferred TF network from human scGLUE[48]-integrated scRNA/ATAC-seq (left) and pig multiome (right) data. Nodes represent TFs, colored by their highly expressed clusters and sized by network centrality. Edges show TF-target interaction strengths (orange: activating; gray: inhibiting). **a**–**c**: scRNA-seq of pancreatic cells from wild-type and *INS*-eGFP pigs. **d**–**f**: Multiome analysis of pancreatic cells from PTF1A-codon-improved-Cre/ROSA-mTmG pigs. Detailed sample information is provided in Supplementary Data 1.

while mainly being expressed in mouse acinar cells. The TF *ETV1* showed enrichment in human delta cells, though in pig and mouse, it was highly expressed in alpha cells.

Conserved cell-type gene expression patterns indicated similarities in higher-order chromatin regulation mediated via cis-regulatory elements (CREs, e.g., enhancers and silencers) to control gene expression[45]. To compare pig and human CREs and upstream

transcriptional regulators of pancreatic cell fate decisions, we additionally performed single-cell 10X multiome sequencing to assess transcriptional changes and chromatin accessibility simultaneously. Samples were collected during the 2° transition of pig development at E45-85, resulting in 29,072 nuclei (Supplementary Data 1). Cluster label transfer from the pig scRNA-seq atlas identified all cell types, except that the MPC cluster had already diminished, while the Epsilon cells

remained undetectable (Supplementary Fig. 5a, b). We further analyzed single-cell transposase-accessible chromatin with sequencing (scATAC-seq) data from human 12 wpc pancreas[30], which had 5592 nuclei. Cell clusters were annotated via label transfer from the human scRNA-seq atlas, and pancreatic epithelial cells were used for further analysis (Supplementary Fig. 5c).

Highly conserved CREs were identified when comparing the sequences between pig and human key lineage regulators, such as the master regulator of endocrinogenesis, *NEUROG3* (Fig. 2d). Further computing cell-type specific motif activities revealed pig-human conserved active TFs towards endocrine and exocrine lineages (Fig. 2e and Supplementary Fig. 5d). Both Acinar and Ductal clusters had active motifs of TFs determining acinar fate differentiation (*NR5A2*, *PTF1A* and *RBPJ*), reflecting the gradual resolution of chromatin and transcription factor profiles in acinar and ductal lineages during terminal differentiation in human and pig. In addition, two MODY genes (*HNF4A* and *HNF4G*) that regulate the growth and function of beta cells, were found active in acinar cells. The Ductal cluster conserved TFs contained Hippo effector genes (*TEAD1* and *TEAD4*) and TFs known to be expressed in the embryonic ductal tree, such as *HNF1A*, *TCF7L2*, *SOX9*, *GLIS3*, and *GATA6*. The human-pig conserved TFs in the EP cluster included known regulators of endocrinogenesis (*NEUROG3*, *RFX3*, *RFX6*, and *NEUROD1*), members of the Nuclear Factor 1 family (*NFIA*, *NFIB*, and *NFIX*), and the intestinal stem cell identity TF *ASCL2*[46]. Endocrine cell clusters also had a large panel of TFs shared between pig and human (*PAX6*, *PDX1*, *MNX1*, *LMX1B*, *PAX4*, *ISL1*, *NKX6-1*, *ARX*, and *NKX2-2*).

Given the conservation of cell-type-specific active motifs between pig and human, we used Pando[47] to construct GRNs using human scGLUE[48]-integrated scRNA/ATAC-seq and pig multiome (joint scRNA/ATAC-seq) pancreas datasets. This enabled detailed examination of relationships between pig-human conserved TFs, their potential target gene expression, and regulatory-site accessibility across cell types and species. The generated GRNs revealed groups of pig-human conserved TFs involved in the differentiation and cell state transition of pancreatic lineages (Fig. 2f and Supplementary Fig. 5e). Ductal-specific TF modules (*HNF1B*, *GLIS3*, and *EHF*), as well as acinar-specific modules (*MECOM*, *XBP1*, and *STAT3*), formed interconnected networks with a set of TFs that showed enrichment in both acinar and ductal lineages (*HES1*, *NR5A2*, *ONECUT1*, *TCF7L2*, *MAFF*, *REST*, and *MEIS1*). Among TF modules linked to endocrine differentiation, *SOX4* (not expressed in pigs) was enriched in both ductal and endocrine lineages in humans. Additionally, *NR3C1*, implicated in regulating islet gene programs and conferring genetic risk of type 2 diabetes[49], was enriched in both the acinar and endocrine lineages in pig, whereas in human, it was only found in endocrine TF modules. Nevertheless, a large group of endocrine TF modules was conserved between pig and human, including *ST18*[50], *DACH1*[51], *PLAGL1*[43,44], and *CDCC88A*[52], genes linked to beta cell mass and function.

## Pig-human conserved endocrinogenesis

To explicate the mechanisms underlying endocrine fate allocation, we focused on the endocrine progenitor clusters stemming from trunk (ductal) cells with low *NEUROG3* expression, progressing towards the cells that clearly diverged into either the alpha or beta cell fate (Fig. 3a–c). To infer developmental trajectories underlying pancreatic lineage acquisition, we applied CellRank with the Palantir Pseudotime kernel[53] to the integrated scRNA-seq atlases to estimate cell states, compute cell-cell transition probabilities, and map cell fates. CellRank correctly predicted the differentiated pancreatic cell clusters as the terminal states (Fig. 3a–c and Supplementary Fig. 6a), providing a reliable transition matrix to infer lineage trajectories in all three species. This revealed distinct differentiation programs in endocrine progenitors of pig and human, opposed to mouse (Fig. 3b). Specifically, at the stage of high *NEUROG3* expression, the progenitors in pig and human already segregated toward either the alpha or beta cell fate.

The presence of *FEV*-expressing endocrine precursors was limited to the beta cell lineage alone. Similarly, the same differentiation pattern was identified by an independent analysis with a subset of the human scRNA-seq atlas data[30]. In contrast, the separation of alpha and beta cell lineages occurred within the *FEV*-expressing endocrine precursors in mice (Fig. 3b and Supplementary Fig. 6b, c).

We next correlated gene expression with CellRank-inferred lineage probabilities and computed putative driver genes of alpha or beta cell fate for each species. Among the orthologous genes mapped across species, pig and human had more overlapping alpha or beta lineage drivers compared to the set shared between mouse and human, regardless of correlation strength (Fig. 3d and Supplementary Data 4). We then identified shared and distinct lineage driver genes with a correlation score >0.7 across species (Supplementary Fig. 6d). To compare expression dynamics of orthologous lineage drivers during endocrine progenitor fate specification, we performed hierarchical clustering in humans to define gene clusters and enriched pathways, then mapped conserved programs in pigs and mice (Fig. 3e, f and Supplementary Data 4). These clusters captured pseudotemporal expression profiles progressing from early endocrine progenitors to fate-committed endocrine cells. For example, in beta cell lineage specification, clusters 3 and 4 showed a conserved expression cascade of Notch signaling-related genes, likely present in early endocrine progenitors. Cluster 2 was enriched with mTOR and RA signaling genes, which showed high expression at both the beginning and end of endocrine progenitor differentiation. Cluster 1 contained genes highly expressed late in differentiation, suggesting their roles in beta cell fate specification. While we observed species-specific differences in extracellular matrix organization, tight junction formation, and cytoskeletal regulation, core features of endocrine differentiation - particularly incretin synthesis and secretion - remained conserved across all species in both beta and alpha cell lineages, demonstrating evolutionary preservation of key differentiation mechanisms.

To evaluate the suitability of pigs as a model of human endocrine development, we performed a comparative analysis of NEUROG3 TF regulatory networks derived from native human and pig pancreas multiome data and experimentally derived targets from human stem cell models. Using Pando-inferred GRNs from human scGLUE-integrated scRNA/ATAC-seq data and pig multiome data, we extracted NEUROG3 networks with significant interactions and module activities (Fig. 3g, h and Supplementary Data 5). Both human and pig NEUROG3 networks contained >100 TFs, including canonical NEUROG3 targets, such as *NEUROG3*, *NEUROD1*, *NKX6-1*, *NKX2-2*, and *MLXIPL*. We then benchmarked these against targets identified through human stem cell models. In our human embryonic stem cell (hESC) model[54,55], we achieved temporal control of NEUROG3 induction using tetracycline to precisely match physiological levels and timing required for endocrine differentiation. These hESCs were differentiated stepwise via endoderm and then foregut into pancreatic progenitors, followed by an 8-h NEUROG3 TF induction. Over 90 high-confidence NEUROG3 downstream TF targets were identified through integrative analysis of time-series mRNA, ATAC and chromatin immunoprecipitation (ChIP) sequencing data (Fig. 3i and Supplementary Data 5). Additionally, we compared these results with NEUROG3 target genes identified using a human induced pluripotent stem cell (hiPSC) model[56], in which an epitope-tagged NEUROG3 was used for cleavage under targets and release using nuclease (CUT&RUN) to identify NEUROG3-bound regions in purified hiPSC-derived EPs (Supplementary Data 5). This revealed 59 conserved NEUROG3 TF targets between native pig and human pancreas GRNs, including 32 targets not detected in either stem cell model, while the stem cell system verified 20 targets absent in the pig in vivo dataset (Fig. 3i and Supplementary Data 5). Several NEUROG3 targets, such as *EHF*, *DACH1*, *ST18*, and *MAFF*, were conserved in all models. Using temporally controlled NEUROG3 TF expression, our hESC model further captured

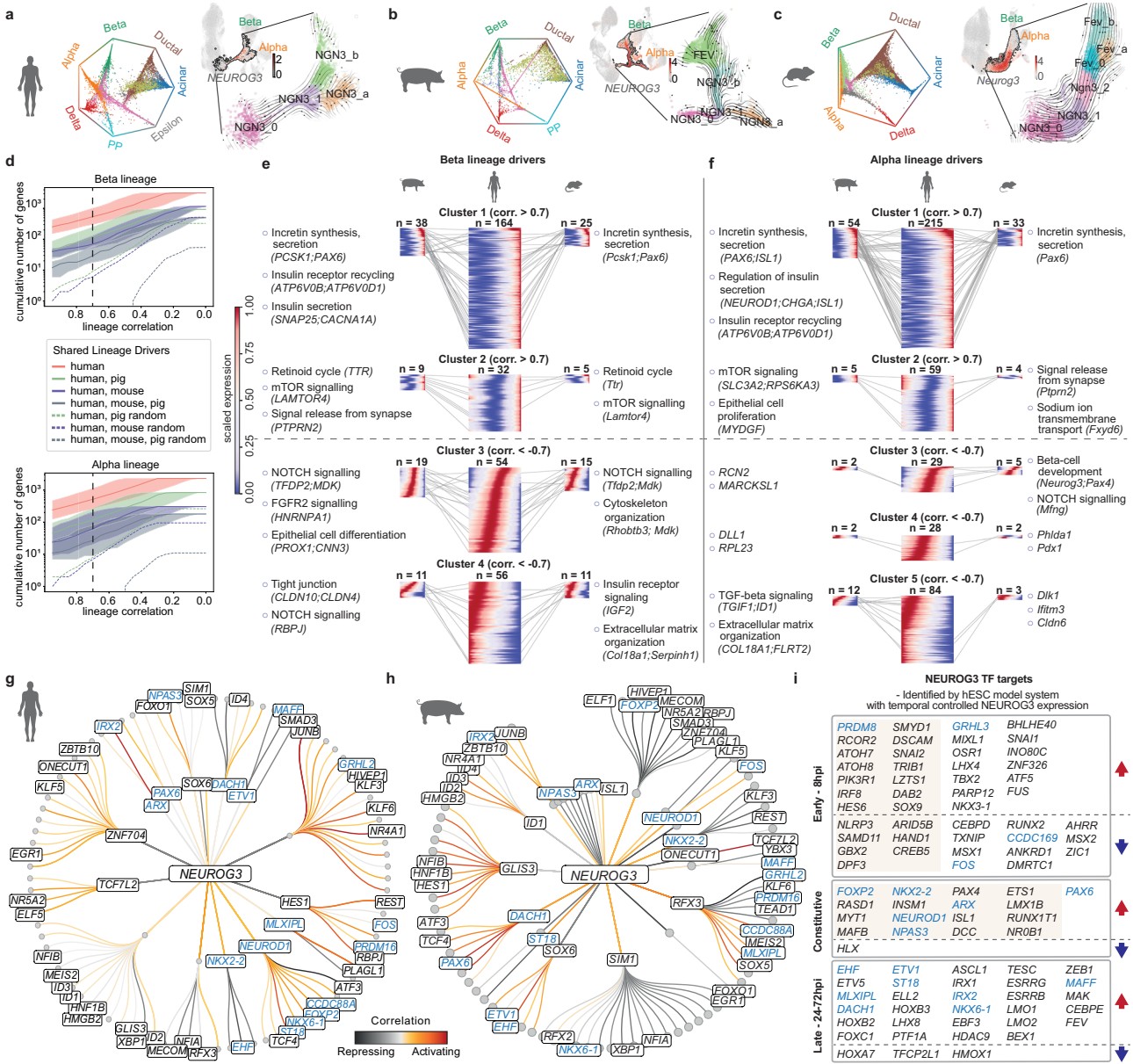

**Fig. 3 | Pig-human conserved endocrine fate allocation. a–c** For each species: circular projection of CellRank-calculated fate probabilities for each cell toward the terminal states (outer labels); and UMAPs detailing endocrine progenitors branching, with the integrated pancreas atlas showing NEUROG3 expression (red) and endocrine progenitor cluster boundaries (black), and an insert showing endocrine progenitor subclusters with overlaid CellRank-inferred trajectories (arrows). **d** Line plots showing the cumulative number of CellRank-derived beta-cell (top) and alpha-cell (bottom) lineage drivers in mouse and pig that overlap with human orthologs, plotted across correlation score thresholds (Supplementary Data 4). Solid lines show significant driver numbers (Benjamini-Hochberg FDR-corrected p-value < 0.05). Shaded regions indicate the number of genes obtained when using the lower and upper bounds of the 95% confidence interval for the corresponding correlation score. **e, f** Heatmaps displaying modeled gene expression patterns for human beta-cell (**e**) and alpha-cell (**f**) lineage driver gene clusters (identified by hierarchical clustering) across pig, human, and mouse along

pseudotemporal trajectories (left to right: 0 → 1). Annotations indicate species-conserved pathways and representative genes for each cluster. (corr., correlation; n, number of conserved genes that are expressed in > 20% of endocrine progenitor subclusters) **g–i** Comparison of NEUROG3 TF targets identified in human/pig pancreas and hESC model (conserved targets in blue). **g, h** Circular GRN graphs showing first- and second-order NEUROG3 targets from human scGLUE-jointed scRNA/ATAC-seq data (**g**) and pig multiomic data (**h**). Nodes represent TFs. Edge color indicates regulatory interaction types (orange, activating; gray, inhibiting); **i** NEUROG3 TF targets in hESC model with inducible NEUROG3 expression. ChIP-seq-identified direct targets are shaded. Differentially expressed TFs comparing cells with/without NEUROG3 expression are indicated by arrows (red, upregulated; blue, downregulated). **a–f** scRNA-seq of pancreatic cells from wild-type and *INS*-eGFP pigs. **h** Multiome analysis of pancreatic cells from *PTF1A*-codon-improved-Cre/ROSA-mTmG pigs. Detailed sample information is provided in Supplementary Data 1.

additional conserved early NEUROG3 targets, such as *PRDM8*, *GRHL3* and *FOS*.

## Developmental origin of beta cell heterogeneity

We and others have previously reported beta cell heterogeneity in terms of proliferation and maturation regulated by the Wnt/planar cell

polarity (PCP) signaling pathway[57,58]. However, the timing and mechanisms underlying the origin of beta cell heterogeneity remain unknown. Intriguingly, two heterogeneous beta cell subpopulations formed during the extended period of pig pancreas development. The Beta,0 cluster consisted of cells appearing at all stages, whereas the Beta,1 cluster emerged mainly during the second wave of

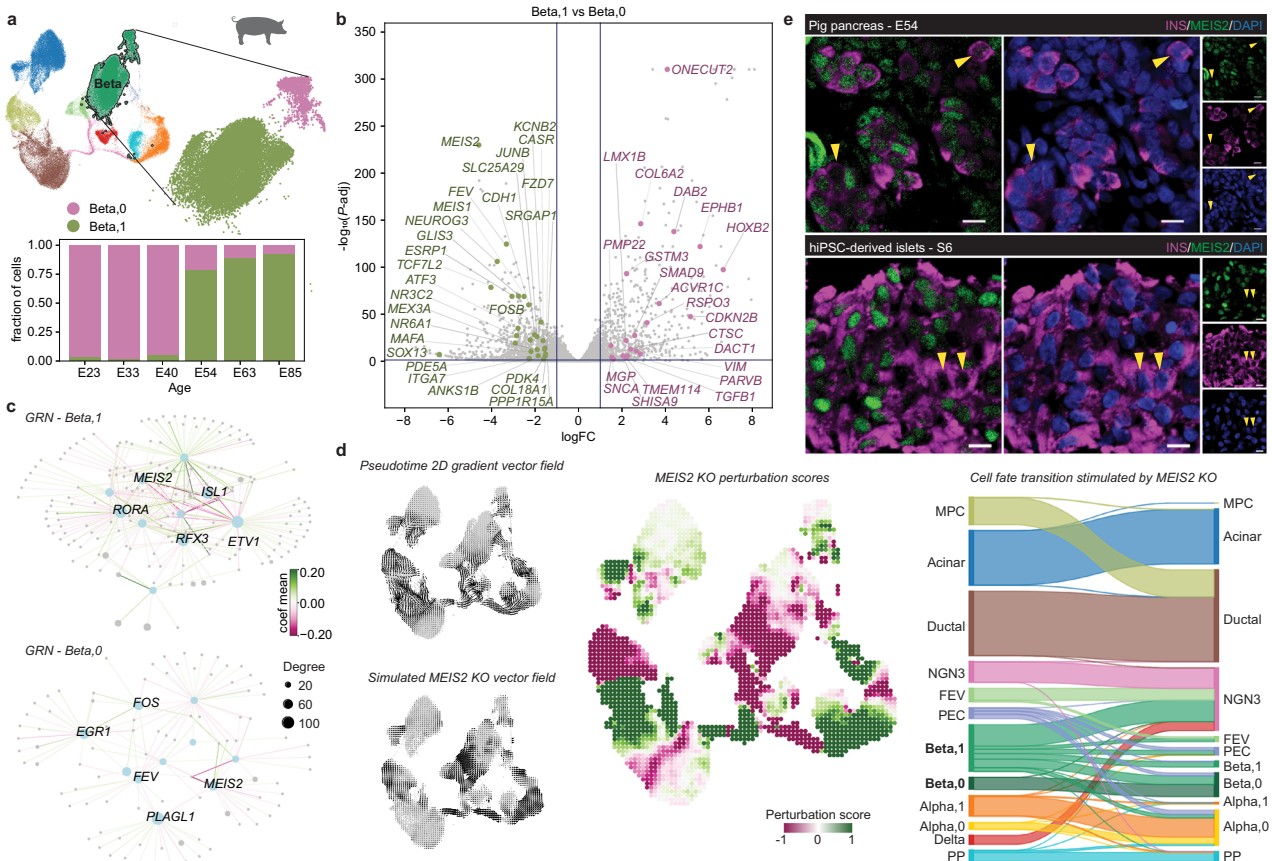

**Fig. 4 | Beta cell heterogeneity arises during pancreas development. a** Detailed view of pig beta cells. (Top) UMAP of pig pancreas atlas with an inset showing a UMAP of the beta cells, colored by subclusters Beta,0 and Beta,1. (Bottom) Distribution of beta cell subclusters at each sampling age. **b** Volcano plot of differentially expressed genes between Beta,0 and Beta,1 (edgeR, FDR-corrected p-value < 0.05; Supplementary Data 6). Genes mentioned in core transcriptional regulatory circuits and identified in pathway analysis are highlighted. **c** CellOracle inferred GRN plots showing networks of the top 5 regulons from Beta,0 and Beta,1 clusters. **d** *MEIS2* in silico KO simulation with CellOracle. (Left) UMAPs overlaid with the Palantir-pseudotime (top) or *MEIS2* KO simulation (bottom) vector fields. (Middle) UMAP colored by *MEIS2* KO perturbation score. (Right) Sankey diagram showing the effect of *MEIS2* KO on cell fate transitions. **e** Immunofluorescence identification of MEIS2 positive and negative beta cells in wild-type pig pancreas and hiPSC-derived islet sections, scale bar 10 μm. Images are representative of 3 pig pancreas samples and 3 independent hiPSC differentiations. **a-d**: scRNA-seq of pancreatic cells from wild-type and *INS*-eGFP pigs. Detailed sample information is provided in Supplementary Data 1.

endocrinogenesis (Fig. 4a). Comparing the two beta subclusters unveiled unique differentially expressed gene sets (Fig. 4b and Supplementary Data 6). Beta,1 cells were enriched in TFs of the reported core transcriptional regulatory circuits for beta cells[59], such as *NEUROG3, FEV, TCF7L2, MEIS1, MEIS2, SOX13, GLIS3, NR3C2*, and *MAFA*. Further gene set enrichment analysis identified active pathways related to epithelial differentiation, extracellular matrix, and cell junction organization (Supplementary Fig. 5a and Supplementary Data 6). In contrast, Beta,0 cells had elevated expression of cell cycle regulators and components of Wnt/planar cell polarity (PCP), TGFβ, and synaptic transmission pathways, suggesting a distinct beta cell phenotype compared to Beta,1 cells.

To gain insight into the regulatory networks shaping the gene expression features of the beta cell subpopulations, we used CellOracle[60] to infer cell-type-specific GRN modules. A custom-assembled base GRN from our pig multiome dataset was applied to construct the GRN configurations in the scRNA-seq data. This resulted in two unique TF networks between Beta,0 and Beta,1 clusters with clearly distinct top 5 regulons (Fig. 4c and Supplementary Data 7). Notably, all these regulons were NEUROG3 targets identified in pig, whereas in Beta,0 cells, only secondary targets were observed (Fig. 3e). *PLAGL1* is a zinc-finger TF implicated in cell-cycle control, ECM organization, and risk of neonatal diabetes[43,44,61–63]. In pig Beta,0 cells, *PLAGL1* formed a network linking genes related to cell migration,

adhesion, and cell-cycle regulation (*TMEM176A, MFAP4, NME2*). In both clusters, *MEIS2* positively targeted beta cell identity genes (*G6PC2, PDX1, CHGA*), albeit interconnecting with different regulons.

*MEIS2* encodes a homeobox TF in the three amino acid loop extension (TALE) family and acts together with PBX and HOX TFs to form dimeric or trimeric complexes to enhance DNA binding specificity and affinity for target gene regulation[64]. MEIS2 was detected in human embryonic[29] and adult beta cells[65] and shown to regulate *PAX6* expression during pancreas development[66]. To understand the role of MEIS2 in beta cell differentiation, we performed in silico perturbation using CellOracle to mimic a *MEIS2* knockout (KO, Fig. 4d). In the *MEIS2* KO simulation, Beta,1 cell differentiation was blocked, with only minimal effects on the Beta,0 cluster. This was further validated by Markov simulation to estimate cell distribution changes, showing that *MEIS2* KO reverted Beta,1 cells to the FEV and NGN3 progenitor states.

Next, we assessed whether MEIS2 could serve as a pig-human conserved marker to distinguish beta cell subpopulations. In the pig scRNA-seq atlas, *MEIS2*+ beta cells mostly appeared in the Beta,1 cluster, while *MEIS2*- beta cells were primarily found in the Beta,0 cluster, both subpopulations sharing characteristics matching their respective beta subclusters (Fig. 5b, c). Remarkably, heterogeneous human beta subpopulations in the scRNA-seq atlas could also be identified based on *MEIS2* expression. Human *MEIS2*+ beta cells resembled pig *MEIS2*+ beta cells and Beta,1 cluster, showing enrichment of genes involved in

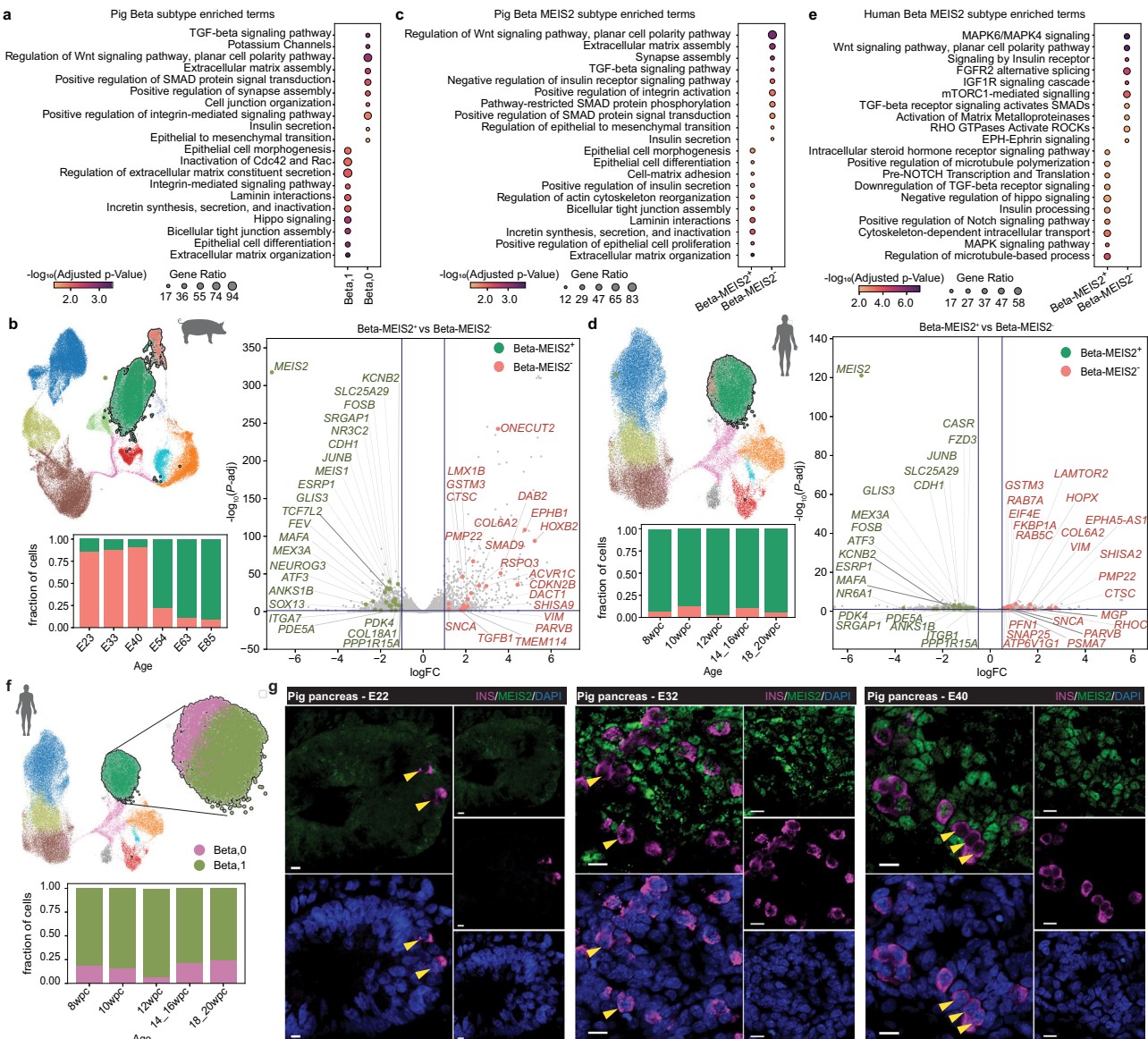

**Fig. 5 | Comparison of beta subpopulations in human and pig. a** Dot plot of enriched pathways identified by Enrichr using the DEGs of pig Beta,0 and Beta,1 clusters. **b** (Left) UMAP showing pig Beta clusters separated according to *MEIS2* mRNA expression with bar plot showing *MEIS2*⁺ and *MEIS2*⁻ beta cell distribution at each sampling age. (Right) volcano plot of differentially expressed genes between pig *MEIS2*⁺ and *MEIS2*⁻ beta cells. **c** Dot plot of enriched pathways identified by Enrichr using the DEGs of pig *MEIS2*⁺ and *MEIS2*⁻ beta cells. **d** (Left) UMAP showing human Beta clusters separated according to *MEIS2* mRNA expression with bar plot showing *MEIS2*⁺ and *MEIS2*⁻ beta cell distribution at the corresponding developmental stage. (Right) volcano plot of differentially expressed genes between human

*MEIS2*⁺ and *MEIS2*⁻ beta cells. **e** Dot plot of enriched pathways identified by Enrichr using the DEGs of human *MEIS2*⁺ and *MEIS2*⁻ beta cells. **f** UMAP showing human Beta clusters separated according to Louvain clustering, with a bar plot showing two beta subcluster distributions at the corresponding developmental stage. **g** Immunofluorescence identification of MEIS2 positive and negative beta cells in wild-type pig pancreas sections from E22, 32, and 40, scale bar 10 μm. Images are representative of 3 pig pancreas samples per time point. All p-values from Enrichr and edgeR analyses are adjusted by Benjamini-Hochberg FDR method (Supplementary Data 6). **a–c**: scRNA-seq of pancreatic cells from wild-type and *INS*-eGFP pigs. Detailed sample information is provided in Supplementary Data 1.

microtubule and cell-matrix organization. Conversely, human *MEIS2*⁻ beta cells had active Wnt/PCP and TGFβ signaling that were observed in pig *MEIS2*⁻ beta cells and Beta,0 cluster (Fig. 5d, e). Louvain clustering further identified a distinct beta subcluster composed of the majority of human *MEIS2*⁻ cells (Fig. 5f). The presence of MEIS2-positive and negative beta cells was confirmed in both pig pancreases (E20-54) and hiPSC-derived islets (Figs. 4e and 5g).

**Embryonic endocrine progenitor heterogeneity**

Having observed distinct beta cell subpopulations emerging during development, we sought to investigate whether these cells may

originate from different endocrine progenitors. We first examined the CellRank-inferred developmental trajectories that faithfully mapped cell fate transitions in the pig scRNA-seq atlas. The NGN3 cluster was predicted to generate major beta cell clusters as expected. Additionally, the PEC cluster was identified as another intermediate state upstream of a beta cell subpopulation (Fig. 6a). To compare the gene expression programs along the trajectories from NGN3 or PEC towards the respective beta cell subtypes, we applied tradeSeq[67] to detect differential gene expression patterns between lineages (Fig. 6b and Supplementary Data 8). Two distinguishable gene clusters were identified comparing lineages NGN3-to-Beta,1 and PEC-to-Beta,1. Gene

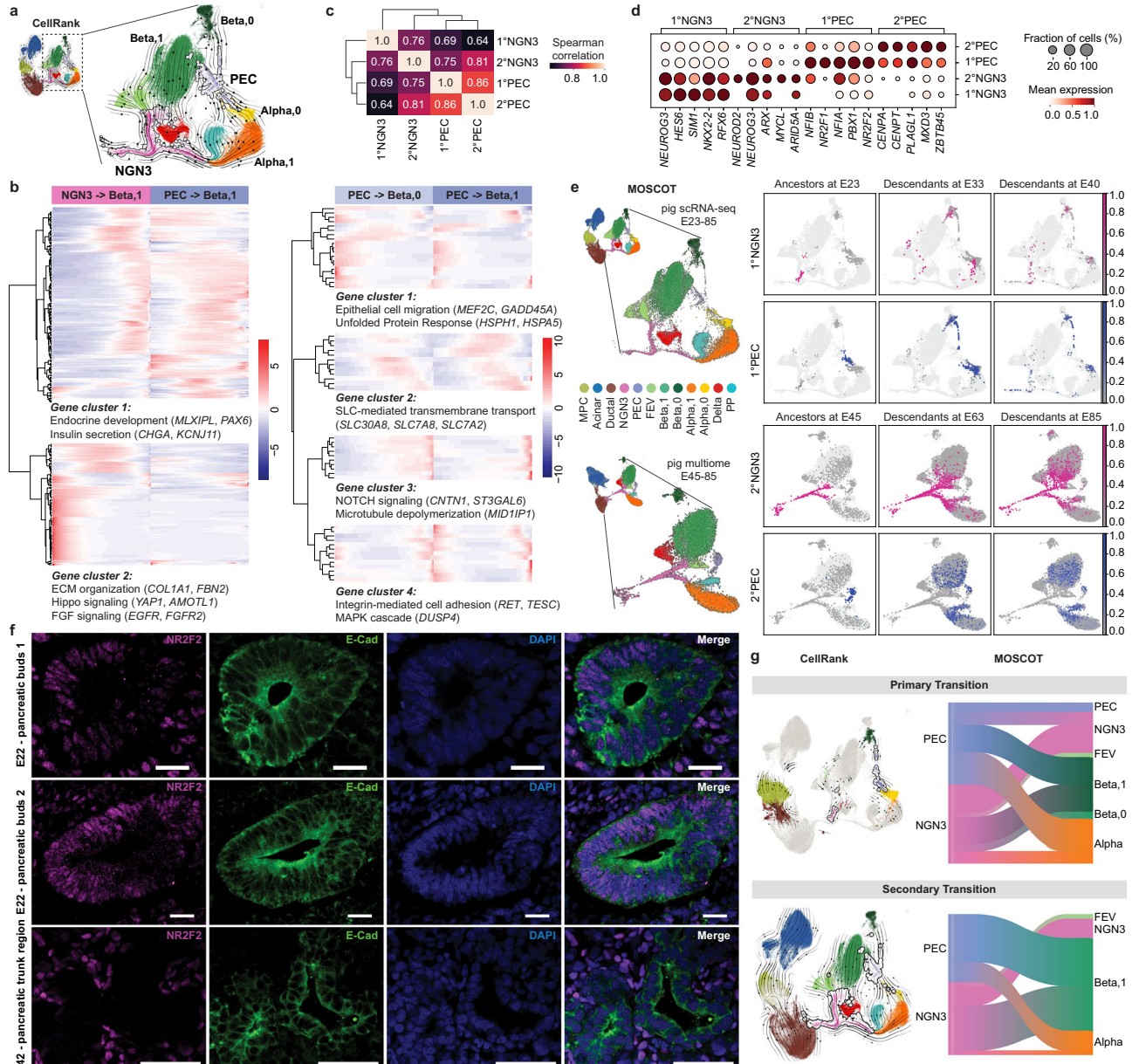

**Fig. 6 | Existence of subtypes of endocrine progenitors during development.**
**a** UMAP showing Cellrank-inferred trajectories of NGN3 and PEC as intermediate states upstream of endocrine cells. **b** Heatmaps showing modeled gene expression patterns along pseudotime for differentially expressed gene groups identified by tradeSeq analysis (Benjamini-Hochberg FDR-adjusted p-value < 0.05; Supplementary Data 8). Comparisons show transcriptional trajectories from NGN3 cluster to beta cell subclusters (left) versus PEC cluster to beta cell subclusters (right). Annotations indicate enriched pathways and signature genes for each gene group. **c** Cluster correlation analysis using Spearman correlation of mean counts per cluster for each highly variable gene ($n = 4000$). **d** Dot plot of the top 5 differentially expressed TFs of cells in 1° transition (E23-33) and 2° transition (E40-85) NGN3 and PEC clusters (edgeR, FDR-corrected p-value < 0.05; Supplementary Data 9). Dot size is relative to the fraction of cells within a cluster expressing the gene. **e** MOSCOT-inferred descendants of NGN3 and PEC cluster from age E23 (using scRNA-seq data, top) and from age E45 (using multiome data, bottom). The left

panel shows the respective UMAP of the endocrine cells from the scRNA-seq (top) and multiome (bottom) atlas used for the MOSCOT calculation.
**f** Immunofluorescence identification of NR2F2+ cells in E22 pancreatic epithelium labeled by E-Cad (top and middle panels). NR2F2 protein is not detected in E42 pancreatic ductal domain (bottom panel). Scale bar 50 μm. Wild-type pig samples are used for this figure. Images are representative of 3 pig pancreas samples per time point. **g** CellRank and MOSCOT (sankey diagrams showing the summarized MOSCOT-inferred ancestors and descendants) prediction of NGN3 and PEC clusters as intermediated states in the 1° transition (E23-33) and 2° transition (E40-85). (Left) UMAPs overlaid with CellRank-inferred trajectories (arrows); (Right) Sankey diagrams showing the summarized MOSCOT-inferred ancestors and descendants in (**e**, **a–d**, **e** (top), **g**): scRNA-seq of pancreatic cells from wild-type and *INS*-eGFP pigs. **e** (bottom) and g: Multiome analysis of pancreatic cells from *PTF1A*-codon-improved-Cre/ROSA-mTmG pigs. Detailed sample information is provided in Supplementary Data 1.

cluster 1 was enriched with genes regulating endocrine development (*MLXIPL*, *PAX6*) and insulin secretion (*CHGA*, *KCNJ11*). It showed overall restricted expression in lineage NGN3-to-Beta,1 as opposed to the extended expression in lineage PEC-to-Beta,1. Gene cluster 2 included genes related with ECM organization, Hippo, and FGF signaling, which

had limited expression in lineage PEC-to-Beta,1. We further compared the two lineages from PEC cluster to either Beta,0 or Beta,1. The identified gene groups comprised known factors involved in beta cell differentiation, such as epithelial cell differentiation, insulin secretion, NOTCH signaling, and integrin-mediated cell adhesion. Surprisingly,

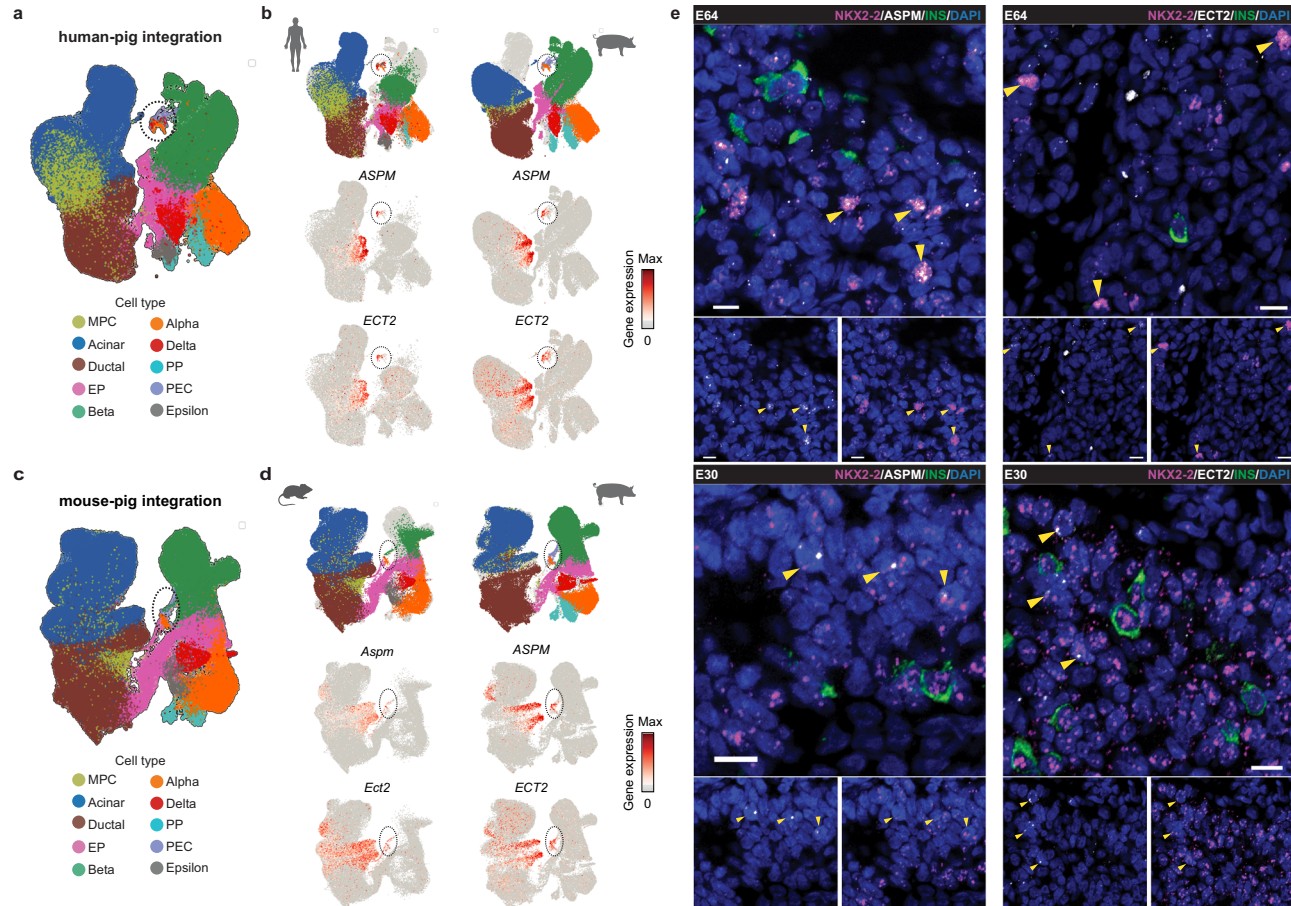

**Fig. 7 | PEC cell confirmation by cross-species integration and RNAscope analysis. a** UMAP of sysVI-integrated human and pig scRNA-seq pancreas atlases. The black circle indicates cells co-clustering with the pig PEC cluster. **b** UMAPs of split views of human (left) and pig (right) data overlayed with the sysVI-integrated embedding, and conserved PEC marker gene (*ASPM* and *ECT2*) expression in each species. **c** UMAP of sysVI-integrated mouse and pig scRNA-seq pancreas atlases. The black circle indicates cells co-clustering with the pig PEC cluster. **d** UMAPs of split views of mouse (left) and pig (right) data overlayed with the sysVI-integrated

embedding, and conserved PEC marker gene (*ASPM* and *ECT2*) expression in each species. **e** Detection of *NKX2-2*+ cells co-expressing *ASPM* (left) or *ECT2* (right) in wild-type pig pancreas samples (E30 and E64) by RNAscope assay. Yellow arrows indicate double-positive cells. Scale bar 10 μm. Images are representative of 3 pig pancreas samples per time point. **a**–**d**: scRNA-seq of pancreatic cells from wild-type and *INS*-eGFP pigs. Detailed sample information is provided in Supplementary Data 1.

these genes showed subtle differences in the gene expression timing and duration, yet without discrete patterns along the two trajectories.

Despite slight deviations from the NGN3 trajectory, PEC-to-Beta gene expression programs favored beta cell differentiation. Hypothesizing that the PEC cluster contains endocrine progenitors, we compared the transcriptional features between PEC and NGN3 clusters. Given the apparent differences in spatial patterning and cell organization of the pancreas during development, we performed correlation analysis of the PEC and NGN3 clusters divided by 1° transition (E23-33) and 2° transition (E40-85), namely following clusters: 1°NGN3, 1°PEC, 2°NGN3 and 2°PEC. Pairwise correlation scores (0.64-1) revealed substantial transcriptional similarity among all four subgroups (Fig. 6c). However, differential expression analysis uncovered distinct patterns in key transcription factors of each cluster (Fig. 6d, Supplementary Fig. 7a, b and Supplementary Data 9). While 1°PEC and 2°PEC showed only marginal expression of canonical endocrine regulators (*NEUROG3*, *NKX2-2*, *NEUROD2*), they were enriched for endocrine-related factors (*NFIA*[68], *PBX1*[69] in 1°PEC; *PLAGL1*[44] in 2°PEC), suggesting a partially active endocrine program. Notably, we identified *NR2F2* as a key transcription factor highly expressed in the 1°PEC cluster, though it was also detected in broader early progenitors and mesenchymal cells (Fig. 6f). *NR2F2* (also known as *COUP-TFII*) encodes a member of the steroid/thyroid hormone superfamily of nuclear receptors. It has been

detected in PDX1+ cells at E11.5 in mice and may play a role in regulating beta cell mass[70,71]. We further confirmed 1°PEC presence in the pancreatic primordium by detecting NR2F2+ cells within the multilayered epithelium of E23 pancreatic buds (Fig. 6f).

To verify the differentiation trajectories linking PEC cluster to beta cell subpopulations, we applied our recently developed Multi-Omics Single-Cell Optimal Transport (MOSCOT)[72] framework, which uses optimal transport theory to reconstruct developmental trajectories across real developmental time points and multiple modalities. Applying MOSCOT to both pig scRNA-seq and multiome time-series atlases recovered the differentiation lineages identified by CellRank, showing that, in addition to NGN3, PEC at both early and late stage was coupled to a subset of endocrine cells as its descendants (Fig. 6e, g).

To determine whether PEC-like cells exist during human and mouse pancreas development, we performed cross-species integration of pig-human and pig-mouse scRNA-seq datasets using sysVI[73]. This machine learning pipeline enables cross-species integration of scRNA-seq datasets while retaining high biological preservation. The sysVI-integrated embedding confirmed correct cell type alignment (Fig. 7a–d), as supported by established pancreas marker gene expression (Supplementary Fig. 7c, d). In both human and mouse datasets, we identified cell populations (initially annotated as diverse endocrine cell types) that co-clustered with pig PEC cells, indicating

transcriptionally shared cellular states (Fig. 7a, c). Notably, mouse PEC-like cells showed connectivity to a subset of cells from the mouse NGN3 cluster, which was not observed in human and pig. We further identified *ASPM* and *ECT2* as cross-species markers, expressed in pig PEC cells and their counterparts in both human and mouse (Fig. 7b, d and Supplementary Data 14). *ASPM* encodes protein abnormal spindle-like microcephaly-associated, which was identified as a novel Wnt and stemness regulator[74]. ECT2 encodes protein epithelial cell transforming 2, a guanine nucleotide exchange factor for Rho-like GTPases[75]. While these markers were also detected in some ductal and acinar cells, PECs showed a clear endocrine identity as evidenced by their strong transcriptional similarity to the NGN3 cluster (Fig. 6c). To validate this population, we selected *NKX2-2*, a pan-endocrine marker, in combination with *ASPM* or *ECT2* to distinguish PECs from ductal and acinar cells. Using RNAscope, we identified *NKX2-2*⁺ cells co-expressing *ASPM* or *ECT2*, confirming their presence in ductal and acinar regions of the pancreas across pig developmental stages (Fig. 7e).

## Discussion

Through cross-species comparative multiomics integrating transcriptomic and chromatin accessibility profiling of pancreas development in mice, pigs, and humans, our work reveals both species-divergent and evolutionarily conserved gene regulatory mechanisms governing pancreatic lineage differentiation. First, the resemblance in developmental tempo of pig and human gestation provides a temporally aligned framework to study extended pancreas organogenesis events that are compressed in the mouse. Second, we observe a pig-human conservation in epigenetic and transcriptional regulation, particularly in the endocrine lineage. The high conservation of transcription factors downstream of NEUROG3 (over 50% between pig and human) suggests a core program for endocrine fate acquisition in larger mammals. This aligns with both the recent human study[30] and the observed similarities in postnatal islet characteristics, fasting C-peptide and glucose levels between pigs and humans[18,21]. By leveraging the temporally resolved single-cell multiomic pig pancreas atlas covering all three trimesters, we identified a unique primed endocrine cell (PEC) population, potentially representing a progenitor state that is distinct from the classic NGN3 endocrine progenitor cluster. Transcriptionally matched PEC-like cells were also identified in human and mouse, with murine PEC-like cells showing a possible link to the NGN3 cluster. Concurrent with pancreas morphogenesis, we discovered that both NGN3 and PEC clusters emerged as heterogeneous populations and were predicted to hold dynamic lineage potential over time. This conserved PEC population is intriguing as it may suggest a potential NEUROG3-independent pathway for endocrinogenesis, which could offer an explanation for the persistence of endocrine cells in some human patients carrying homozygous NEUROG3 mutations[76,77]. However, definitive validation of this pathway and the differentiation capacity of PECs remains a crucial goal for future work. We further identified heterogeneous beta cells, suggesting the acquisition of islet cell heterogeneity may originate in embryonic development. In summary, our study presents the pig as a valuable complementary model to existing systems and enhances the understanding of in vivo pancreas development through a comprehensive multimodal comparison across species. These resources can be harnessed to refine stem cell and organoid models, offering unique opportunities to 1) address open questions in human biology and disease, and 2) bridge translational gaps from rodents to large animals (pigs) and ultimately to humans.

## Methods

### Pig pancreas collection

Pigs were housed at the designated pathogen-free pig facilities in LMU or TUM. All animals received standard diet and water *ad libitum* as well as standard vaccination. After hormonal estrus cycle synchronization,

pigs in heat were artificially inseminated or mated and ultrasonic confirmation of pregnancy was performed 21 days post insemination. At selected gestation stages, pregnant pigs were euthanized, and fetuses were collected. Pancreases were extracted and used for further analysis.

### Immunofluorescence analysis

Pig pancreases or hiPSC-derived islets were fixed in pre-chilled 4% paraformaldehyde overnight (pancreas) or for 30 min (islets) then dehydrated in a progressive sucrose gradient at 4 °C. Samples were embedded in tissue freezing medium (Leica Biosystems), sectioned at 12 μm, and mounted onto Superfrost® Plus slides (Thermo Fisher Scientific) and stored at -80 °C.

**Immunostaining.** Sections were washed with PBS for 30 min, permeabilized in 0.1% Triton X-100 (Sigma-Aldrich) in 0.1 M Glycine (Sigma-Aldrich) for 30 min, blocked in blocking solution (PBS + 0.1% Tween-20 + 10% FCS + 0.1% BSA + 3% donkey serum) for 1 h at room temperature (RT), and incubated with primary antibodies diluted in blocking solution (Supplementary Data 10) overnight at 4 °C. Sections were washed in PBS for 15 min twice, incubated with secondary antibodies diluted in blocking solution (Supplementary Data 10) for 3 h at 4 °C, counterstained with DAPI for 20 min at RT, and mounted with the ProLong™ Diamond Antifade Mountant (Thermo Fisher Scientific).

**Integrated mRNA and protein co-detection assay.** Pig pancreas sections were used for mRNA and protein co-detection assay according to the RNAscope® Multiplex Fluorescent v2 Assay combined with Immunofluorescence - Integrated Co-Detection Workflow (ACD, Biotechne). Briefly, sections were washed in PBS, baked at 60 °C, and dehydrated in a progressive ethanol gradient. Sections were treated with hydrogen peroxide and 1x Co-Detection Target Retrieval solution and incubated with primary antibody (Supplementary Data 10) diluted in Co-Detection Antibody Diluent overnight at 4 °C. Sections were fixed in 10% neutral buffered formalin for 30 min at RT and treated with RNAscope® Protease III. Sections were then hybridized with probes against pig *NEUROG3* (ACD, Cat#498781), NKX2-2 (Cat#1570601-C1), ASPM (Cat#1734071-C3), and ECT2 (Cat#1734091-C3) for 2 h at 40 °C and signals developed following the RNAscope® Multiplex Fluorescent Reagent Kit v2 Manual. Sections were incubated with secondary antibodies for 30 min at RT, counterstained with DAPI for 20 min at RT, and mounted with the ProLong™ Diamond Antifade Mountant (ThermoFisher). All slides were stored at 4 °C until imaging by confocal microscope (ZEISS LSM 880 with Airyscan). Images were acquired and processed using Zeiss Zen 2.3 lite (Blue edition, ZEISS).

### Improved genome annotation for Sus scrofa

Pig pancreas scRNA-seq data was previously aligned using Ensembl *Sus scrofa* gene annotation (version 94), however showed incomplete gene annotation missing certain genes of interest (e.g. *MAFA*, *FEV*, *PTF1A*). Our analysis found reads downstream of the gene bodies which were not covered by the annotation. Beiki et al.[82] generated an improved pig genome annotation (iso-seq annotation) by integrating poly(A) selected single-molecule long-read isoform sequencing (Iso-seq) and Illumina (short read) RNA sequencing (RNA-seq). Therefore, we further improved the annotation by combining iso-seq annotation with the Ensembl annotation version 101. We annotated the transcripts of iso-seq annotation using gffcompare (v0.12.1)[83] with referencing the annotation from Ensembl version 101. The Ensembl annotation was filtered by removing "pseudogene" and "processed_pseudogene" biotypes genes. We added the associated same strand transcripts from the iso-seq annotation to the Ensembl annotation and extended the gene bodies if the added transcripts required it. If a transcript was added to a gene, the gene ID was altered by adding the suffix -iso. For each gene, we then added an additional "extension gene" downstream of the gene

body. The length of the extension gene was defined by the maximum possible region of 1-10 kb (using 1 kb steps) that does not intersect with any other same strand gene. The extension genes were named by the name of the original genes and adding the suffix -ext'x'kb, whereby 'x' corresponds to the integer length of the addition from 1-10. Final gene counts were obtained by summing the -iso and -ext'x'kb (if available) versions of each gene. Supplementary Data 11 shows UMI counts for 37 genes of interest before ("ensembl") and after ("iso") gene extension as examples.

### Single-cell sequencing

**Biological replication.** For next-generation sequencing experiments, a minimum of three biological replicates (individual embryos/fetuses per pregnancy) at each developmental stage were sampled where possible. Modified approaches were used for two exceptions: At E40, only two embryos were available for analysis; At E22/23 and E33, pancreatic tissues were extremely small in size. To obtain sufficient cell numbers for reliable single-cell sequencing, all available pancreases at each age were combined for processing, as summarized in Supplementary Data 1 and Supplementary Fig. 2a, c.

**Single cell suspension preparation.** Freshly dissected pancreases were minced into fine pieces and digested with collagenase V in HBSS with Ca/Mg (0.5 mg/mL for E20-50, 1 mg/mL for E60-85, Sigma-Aldrich) for 5 min followed by dissociation with TrypLE™ Express (Gibco) for 10–15 min at 37 °C. Cell suspension was filtered through a 40 μm cell strainer.

**Pancreatic epithelial cell enrichment.** This procedure was not applied to E22/23 and E33 samples due to their small size; all cells from these samples were used for scRNA-seq without enrichment. Pancreases from transgenic reporter pigs and wild-type littermates at E40-85 were used to collect pancreatic epithelial cells. For enrichment, cell suspension was stained with EpCAM-PE-Cy7 antibody (1:200, Invitrogen) in 1% BSA + PBS for 30 min at 4 °C. Cells were then washed and filtered into a tube through a 35 μm filter. 7-AAD (Invitrogen) or Sytox blue (Invitrogen) was used to distinguish dead cells. The resulting cell suspension was loaded onto a FACSAria III (BD) for sorting. The transgenic reporter pigs were used for specific cell-type enrichment: 1) *INS*-eGFP pigs[78]: beta cells were enriched via insulin promoter-driven GFP expression. 2) *PTF1A*-codon-improved-Cre[79] x *ROSA*-mTmG[80] pigs: *PTF1A*-codon-improved-Cre/*ROSA*-mTmG embryos were identified with epifluorescent microscope by GFP signal. The pancreases from these embryos were used to enrich pancreatic epithelial cells, particularly acinar cell populations, with GFP signal. Supplementary Data 1 provided an overview of all samples obtained through different enrichment strategies. Supplementary Fig. 2c provided a summary of pancreas number and cell counts of identified cell types in each sample. Supplementary Fig. 8 provided representative gating strategies.

**scRNA.** Single-cell suspensions were processed for scRNA-seq with a targeted cell recovery of 10,000. 10X Genomics' Single Cell Gene Expression protocols were followed according to the manufacturer's specifications and guidelines. Libraries were pooled and sequenced by a HiSeq4000 or NovaSeq 6000 platform following the recommendations from 10X Genomics. With CellRanger pipeline (v3.1.0), samples were demultiplexed to produce a pair of FASTQ files for each sample. Reads containing sequence information were aligned to the improved pig genome annotation and pre-processed for downstream analyses.

**Multiome (scRNA/ATAC).** For nuclei isolation and library construction, a low-input nuclei isolation protocol adapted from 10X Genomics was performed. In brief, sorted cells were washed once with 1 mL PBS + 1% BSA, counted, centrifuged, and the supernatant was aspirated. Subsequently, the washed cell pellet was resuspended in chilled

lysis buffer with 0.5x detergent concentration (50 μL per sample) and placed on ice for 5 min. Then wash buffer (500 μL per sample) was added and nuclei were centrifuged. To gradually change from wash to diluted nuclei buffer, cells were washed once in a 1:1 mixture of wash buffer and diluted nuclei buffer and subsequently one with pure diluted nuclei buffer. The washed isolated nuclei were then resuspended in 7-10 μL diluted nuclei buffer and were directly added to the transposition reaction after quality control and counting. In all following steps, 10X Genomics' Single Cell Multiome ATAC and gene-expression protocols were followed according to the manufacturer's specifications and guidelines. The final libraries were sequenced on the Illumina NovaSeq 6000 platform following the recommendations from 10X Genomics. Raw reads were aligned to the improved pig genome annotation and pre-processed using the 10X Genomics Cell-RangerARC pipeline (v 2.0.0) for downstream analyses.

### Single-cell data analysis

**Preprocessing of 10X scRNA-seq raw data.** Pig pancreas samples (Supplementary Data 1) and published datasets of human and mouse[30,35–41] were preprocessed using Scanpy[84] (v1.8.2).Filtering of low-quality cellsEach sample was assessed using Scanpy's quality control measures and sample-specific minimum number of genes per cell, minimum number of counts per cell, and maximum number of counts per cell were set to filter out low-quality cells (Supplementary Data 12). In addition, all cells with a mitochondrial fraction > 0.15 were excluded, as well as all genes that were expressed in less than 20 cells. Read counts and gene counts across clusters of pig scRNA-seq data were shown in Supplementary Data 13. The filtered gene matrices from Goncalves et al.[37] were not filtered.NormalizationGene counts were normalized using Scran[85] (v1.22.1) for data from each lab separately. For this, we first performed a total counts normalization of each cell counts to 1,000,000, then performed a log transformation using natural log and pseudocount 1, further calculated a neighborhood graph using the first 50 principal components and number of neighbors k = 15. Clusters were obtained using louvain clustering[86] with resolution r = 0.5. We then used Scran to estimate size factors with the louvain clusters as input clusters and minimum mean average count of genes to be used for normalization set to 0.1. The size factors were then used for normalization of raw gene counts (summed -iso and -ext'x'kb gene counts for pig samples). For downstream integration, log-transformed counts using natural log and pseudocount 1 were calculated.Ambient gene calculationAmbient genes were estimated based on expression in empty droplets using DropletUtils[87] (v1.14.2). Genes with an ambient expression score larger than 0.005 were considered ambiently expressed genes. Ambient genes were generalized to datasets where raw data was not available.Highly variable gene calculationHighly variable genes (HVGs) were calculated per batch[88] to select HVGs unaffected by batch variance. Per-batch HVGs were obtained with Scanpy, using the CellRanger[89] flavor, and were ranked first by the number of batches in which the genes were highly variable and second by the mean dispersion across batches. Finally, the top genes in this ranking were selected as highly variable genes.IntegrationPig samples were integrated with Scanorama[90] (v1.7.1) using 8,000 HVGs (excluding ambient genes), resulting in a 100-dimensional latent embedding. Human and mouse samples were each integrated with scVI[91] (v0.16.1) using 2,000 HVGs, resulting in a 20-dimensional latent embedding.-Clustering and annotationClustering was performed on the k-nearest neighbor (KNN) graph (k = 15) calculated from the integrated embedding using louvain clustering[86] (leiden[92] for human and mouse data) with resolution 1. The pig integrated embedding was reduced to 50 principal components before calculating the neighborhood graph. Clusters were annotated using differentially expressed marker genes. For some analysis several clusters (NGN3, Beta) were subclustered using a higher clustering resolution.Re-normalization after integrationScran normalization might be based on batch-specific clusters when a strong

batch effect is present, leading to non-comparable counts across samples. We ran a check on all species datasets by visualizing the mean number of counts per cluster (based on integrated embedding) and sample before and after normalization. For human and mouse samples we observed strong sample-specific counts after normalization. To obtain comparable counts across samples, we re-normalized these samples using Scran and the clusters obtained from the integrated embedding.Doublet detectionTo obtain robust doublet estimates, we used a combination of scrublet[93] (v0.2.3), DoubletDetection[94] (v4.2), scds[95] (v1.10.0), scDblFinder[96] (v1.11.4), DoubletFinder[97] (v2.0.3) (default parameters, expected doublet rate 0.8) to detect doublets. Cells consistently detected by three or more methods as doublets were excluded from further analysis. In addition, clusters with a doublet frequency larger than 70% were entirely excluded.

**Reanalysis of human fetal scATAC-seq dataset.** The 12wpc human fetal scATAC-seq data[30] were read into Signac[98] for preprocessing. Peaks from standard chromosomes and additionally called using MACS2 were used. Gene annotation from EnsDb (EnsDb.Hsapiens.v86) was added. Quality control metrics were computed to filter low quality cells (Supplementary Data 12). Data was normalized by term frequency-inverse document frequency (TF-IDF) normalization. Dimension reduction was done by running singular value decomposition (SVD) on the TD-IDF matrix, using the peaks selected by the function FindTopFeatures. Graph-based clustering and non-linear dimension reduction for visualization was performed on the KNN graph (k = 30) calculated from the low-dimensional embedding using SLM[86] algorithm. Gene activity matrix was created by the GeneActivity function. Doublets were called by scDblFinder[96] and excluded for further analysis. For cell cluster label transfer, we first extracted the 12wpc data subset from the integrated human scRNA-seq data. The scATAC-seq data gene activity was used as an approximation of a gene expression matrix and was integrated with the 12wpc scRNA-seq data following the standard scANVI[99] (v0.20.3) workflow to enable label transfer. Cicero was used to compute pairwise co-accessibility scores for each peak, which were further grouped into cis-co-accessible networks. The co-accessible links along with DNA accessibility information were visualized by CoveragePlot.

**Integration of human fetal scRNA-seq and scATAC-seq data.** The unmatched modalities were integrated using GLUE[48] v0.3.2. The RNA modality input, i.e. the 12wpc data subset was extracted from the integrated human scRNA-seq data. The ATAC modality input was processed as described in section "*Reanalysis of human fetal scATAC-seq dataset*". We then constructed a guidance graph that contains omics features as nodes (i.e., genes for scRNA-seq, and peaks for scATAC-seq) and prior regulatory interactions as edges. We used the default implementation that links an ATAC peak to a gene if it overlaps either the gene body or promoter region. This graph was utilized by GLUE to orient the multi-omics alignment. To match cells from both modalities, we performed minimum cost maximum flow bipartite matching on the joint embedding derived from GLUE[47,100]. The cost graph was inferred using get_cost_knn_graph with knn_k = 15, null_cost_percentile = 99 and capacity_method = 'uniform'. Using the bipartite matches, we matched each ATAC cell to an RNA cell. In cases where no ATAC match was found for an RNA cell, only RNA information was used. The latent vector of the cell was calculated as the average latent vector of the matched cells. Gene activities were further denoised with MAGIC[101] by smoothing over nearby cells in the joint embedding as proposed and benchmarked in ArchR[102]. The Python implementation of magic (v3.0.0) was used to smooth gene activities over the k-nearest neighbors graph of the joint embedding with k = 15 neighbors, decay = 1 and k-nearest neighbors autotune parameter ka = 4. The integrated and imputed dataset was used for gene regulatory network construction.

**Preprocessing of pig 10X multiome raw data.** Multiome data was preprocessed similarly to scRNA-seq data as described above using Scanpy (v1.9.1) and Muon[103] (v0.1.2). After summing the -iso and -ext'x'kb (if available) counts of each gene to generate final counts, DropletUtils[87] (v1.14.2) was used with default parameters to estimate ambient gene expression probabilities.

**Filtering of low-quality cells.** Each sample was assessed using standard quality control measures and sample-specific maximum mitochondrial gene fraction, minimum number of genes per cell, minimum number of counts per cell, and maximum number of counts per cell were set to filter out low quality cells (Supplementary Data 12). To further filter out cells with low ATAC-seq quality, Muon was used to calculate ATAC-specific quality metrices. Sample-specific thresholds were identified for minimum and maximum number of counts, minimum and maximum TSS enrichment score as well as minimum and maximum nucleosome signal (Supplementary Data 12).

**Doublet detection.** Similar to scRNA-seq, we used a combination of scrublet[93] (v0.2.3), DoubletDetection[94] (v4.2), scds[95] (v1.10.0), scDblFinder[96] (v1.11.4), DoubletFinder[97] (v2.0.3) (default parameters, expected doublet rate 0.8) and SOLO[104] (as implemented in scvi-tools v0.19.0) to detect doublets based on the gene expression modality. In addition, scDblFinder and its implementation of AMULET[105] were used to identify doublets on the ATAC-seq modality. Cells consistently detected by three or more methods as doublets were excluded from further analysis.

**Generation and quantification of common peak set.** To merge the ATAC-seq data from individual samples, we followed the respective vignette on the Signac97 website. In brief, peaks from all samples were merged using the "reduce" function of the GenomicRanges (v1.46.1) package and only peaks on standard chromosomes were kept. Next, for each sample, fragment counts were determined using Signac and stored, together with gene expression data in a Seurat object, which were subsequently merged into a single object.

**Normalization of ATAC-seq counts.** Signac was used to run TF-IDF normalization on ATAC-seq counts with default parameters. TF-IDF normalized count matrix was then imported into Muon.

**Normalization of gene expression counts.** Prior to normalization, data from individual samples was merged. SCTransform (v0.3.3) was used for normalization using settings vst.flavor = "v2" and clip.range=c(-sqrt(n), sqrt(n)),where n ($n = 33898$) represented the number of cells.

**Highly variable genes.** The top 4000 highly variable genes were identified using the devianceFeatureSelection function from the scry package[106] (v1.6.0) with default parameters.

**Label transfer from scRNA-seq.** To ensure consistent cell type labels between scRNA-seq and Multiome data, we employed a k-nearest neighbors classifier to transfer cell type annotations from the scRNA-seq reference. We first trained an scVI model (scvi-tools v 0.20.0), with parameters n_hidden=1024, n_latent=50, n_layers=2, gene_likelihood = 'nb', dispersion = 'gene-batch', sample names as batch key, and the technology (i.e. scRNA-seq and Multiome) as categorial covariate, to generate a shared latent space for the scRNA-seq reference and the Multiome gene expression data. Next, we used the k-nearest neighbors classifier (k = 5), as implemented in scikit learn (v0.24.2), to predict cell type labels in the shared latent space.

**Integration.** To integrate the gene expression modality of the different Multiome samples, we used harmonypy[107] (v0.0.9) with sample names

as batch key. To integrate the chromatin accessibility modality, we used PoissonVI[108], with parameters n_hidden=1024, n_latent=50, n_layers=3, sample names as batch key, transferred labels as labels key and the developmental stage as categorial covariate.

**Clustering and annotation.** Clustering was performed on the k-nearest neighbor (KNN) graph (k = 21, metric = 'minkowski') calculated from the harmonypy integrated embedding using leiden[92] clustering with resolution 2. To further separate subtypes of endocrine progenitors, the respective clusters were subclustered with a resolution of 1. The resulting clusters were then annotated to match the transferred cell type labels.

**Differential gene expression analysis using edgeR.** We calculated differentially expressed genes between cell clusters by pseudo-bulking samples with edgeR[109] (v4.0.16). We calculated pseudo-bulk expression by summing normalized raw counts of cells from one sample and cell type. Pseudo-bulks with fewer than 30 cells were excluded. To ensure that ambiently expressed genes were not erroneously predicted as differentially expressed genes, we balanced the pseudo-bulks on the sample level. For differential testing, we modeled gene expression using a generalized linear model[110] with cell type and sample as covariates. Significantly differentially expressed genes were identified using a likelihood-ratio test for the coefficients of interest ($q < 0.05$, corrected for multiple testing with the Benjamini-Hochberg method[111] at alpha=0.05). Gene set enrichment analysis was done with Erichr[112] to identify pathway enrichment signatures of the respective cell cluster.

**Cross-species comparison using cluster correlation.** We compared cell-type gene expression profiles across species. For this, we calculated the mean normalized gene count per cluster. Then, we mapped pig and mouse genes to human gene symbols using orthologues from Ensembl BioMart[113]. If there were multiple genes mapping to one human gene, we used the summed mean normalized gene counts. For the correlation, we considered only the intersection of the top 4000 highly variable genes from all three species, resulting in 851 genes. We used Spearman's rank correlation[114] to compare gene expression values for each cluster across species.

**Trajectory inference of scRNA-seq data with CellRank.** We estimated developmental trajectories and cell fates using CellRank[53] (v1.5.1). For this, we estimated a pseudo-time for every cell using Palantir[115] (v1.0.1), with a highest *PDX1*-expressing MPC cell as root cell from E22 for pig, 49dpc for human and e9.75 for mouse, respectively. Using the CellRank pseudo-time kernel, we calculated terminal states for all clusters and endocrine progenitor clusters (NGN3 and FEV clusters in pig and mouse data, EP in human data).

**CellRank lineage driver estimation and cross-species comparisons.** To compare endocrine development across species, we computed putative lineage drivers by calculating Pearson's correlation of each gene with CellRank fate probabilities for each terminal state. We mapped pig and mouse genes to human gene symbols using orthologues from Ensembl BioMart[113]. If there were multiple genes mapping to one human gene, we used the maximum absolute lineage correlation score. We then only considered genes that were present in all three species and had orthologues, resulting in 12,437 genes. To compare lineage correlation scores, we scaled the scores per species by the maximum of the 0.01 and 0.99 quantile and clip values to -1 and 1. Finally, we calculated the lineage driver genes (Benjamini-Hochberg FDR-corrected $p > 0.05$, scaled correlation>0.7) of this mapped subset for every species and lineage and compared overlaps across species.

**Gene module analysis of CellRank lineage drivers.** To compare gene expression dynamics during alpha/beta cell development in mouse, pig and human, we first extracted orthologous alpha/beta lineage drivers expressed in at least 20% human cells and performed hierarchical clustering with AgglomerativeClustering from scikit-learn[116] to identify gene groups for lineage drivers with positive or negative correlation, respectively. Gene set enrichment analysis was performed with GSEApy enrichr module[112] to identify pathway enrichment signatures for each identified gene group. The expression patterns of these gene groups were analyzed in pig and mouse data.

**Differential gene expression analysis between lineages by tradeSeq.** We used tradeSeq116[67] (v1.14.0) to calculate differentially expressed genes between lineages, i.e., NGN3-to-Beta,0 versus (vs) PEC-to-Beta,0; PEC-to-Beta,0 vs PEC-to-Beta,1. CellRank computed trajectories for these lineages was extracted and used for tradeSeq downstream analysis. We fitted a negative binomial generalized additive model (NB-GAM) using tradeSeq for each of the top 2000 highly variable genes and each lineage. We then identified genes with significantly different expression patterns between lineages using the PatternTest function. To exclude the genes that were already differentially expressed at the initial or terminal states, we inverted the gene rank from the edgeR differential expression (DE) analyses of NGN3 vs PEC and Beta,0 vs Beta,1. We further scored each gene by combining the inverted DE rank and the rank in the PatternTest results (transientScore). This allowed us to identify genes with similar expression at the initial or terminal state but showed significantly different expression pattern along the lineage ($q < 0.05$, corrected for multiple testing with the Benjamini-Hochberg method[111] at alpha=0.05). Gene set enrichment analysis was performed with GSEApy enrichr module[112] to identify pathway enrichment signatures for each identified gene group.

**Cross-species integration by sysVI.** To integrate scRNA-seq dataset from different species, we used sysVI[73] model that combines machine learning with conditional variational autoencoders (cVAE) for integrating datasets with substantial batch effects while better preserving biological variation. We subset the pig and human datasets to the intersection of top orthologous HVGs (2535 genes) as recommended. The two datasets were concatenated with "species" defined as batch_key covariate and "samples" defined as categorical_covariate_keys covariate. The default implementation of sysVI with multimodal variational mixture of posteriors prior (VampPrior) combined with latent cycle-consistency loss was used for the integration.

**Gene regulatory network inference using CellOracle.** Using CellOracle[60] (v0.12.0), we first constructed a base GRN using the scATAC part of the multiome data. The co-accessible peak information was extracted to generate the active gene regulatory element data, which contained the open accessible genomic regions and cis-regulatory connection data. We then annotated transcription start sites (TSS) to generate the active promoter/enhancer genomic region data. These data were integrated and peaks with weak co-accessibility scores removed, resulting in the final pig base GRN. Pig scRNA-seq data was reduced to 25,000 cells with 3028 genes (top 2000 HVGs + all TFs). An Oracle object was built by combining the gene expression counts, clustering information, CellRank trajectory with the base GRN. After KNN imputation, cluster-specific GRN for all clusters was calculated with the get_links function. To remove the weak edges and insignificant edges, we filtered the network edges by keeping the top 2000 edges ranked by edge strength with a p-value < 0.001 before network structure analysis. Network (centrality) scores were calculated using the links.get_network_score function. The top genes ranked by betweenness centrality were selected for Beta,0 and Beta,1 cluster to

plot the GRNs with NetworkX[117]. To simulate *MEIS2* KO in silico, we first used the Oracle object and the filtered GRNs of all clusters to make the regression models (a regularized linear machine-learning model) for simulation. *MEIS2* expression was set to 0, and the global future gene expression shift after perturbation was then calculated.

**Gene regulatory network inference using Pando**
**Metacells.** To overcome the sparsity of single-cell data, we used SEACells[118] (v0.3.2) with default parameters to identify metacells ($n = 387$), representing cell states in the integrated gene expression latent space.

**Peak to gene linking.** To identify putative regulatory elements, we used the Signac (v1.9.0) LinkPeaks function with default parameters to calculate the correlation between chromatin accessibility and gene expression of nearby highly variable genes.

**GRN inference.** We then used all peaks with significant links to construct a GRN using Pando[47] (v1.0.3) for both pig multiome data and human integrated scRNA/ATAC-seq data. In brief, we used the motif collection and find_motifs function from the Pando package to identify transcription factor motifs within the peaks. We then inferred the GRN considering only highly variable transcription factors found in the motif collection and peaks within $1 \times 10^6$ bp around their TSS, using the following parameters: peak_to_gene_method = 'Signac', aggregate_peaks_col = 'SEACell', tf_cor = 0.05, method = 'xgb'. Next, we constructed transcription factor modules within the GRN using the find_modules function with a p-value threshold of 0.05, a $R^2$ threshold of 0.15 (0.05 for human data), a minimum number of variables of 50 (10 for human data) and a minimum number of genes per module of 5. The GRN was visualized as a UMAP embedding of the TFs based on co-expression and regulatory relationship as measured by the inferred coefficients. Nodes were sized by the PageRank centrality of each TF and colored according to the enrichment of TF expression. Coverage tracks and peak-to-gene links were visualized using Signac. UCSC lift-Over tool (http://genome.ucsc.edu)[119] was used to lift link coordinated from pig to human reference genome. ALRA[120] was used to impute pig gene expression and the calculated values were shown as violin plots next to the coverage plots.

**Motif activity analysis with ChromVAR.** We computed motif activities for human and pig scATAC-seq data using chromVAR[121] (v1.24.0). We first identified motif matches of the human_pwns_v2 motif collection from the chromVARmotifs package using Signac. Motif class information was derived from CIS-BP Database Build 2.00. Next, we used the RunChromVAR function with default parameters to calculate per-cell motif activity scores. We then tested for differential activity scores between cell types using the FindAllMarkers function with mean.fxn = rowMeans, to compute the average difference in z-scores. Motifs with an adjusted p-value < 0.01 and an average difference > 1 were considered differentially active. To further select meaningful motifs for plotting in the heatmaps in the main figures, we computed the Pearson correlation between transcription factor gene expression z-scores and chromVAR motif z-scores in the pig multiome data and kept only motifs/transcription factors with a correlation coefficient > 0.1. Heatmaps show the chromVAR z-scores stored in the data slot of the chromVAR assay.

**Trajectory inference of scRNA-seq and Multiome data with MOSCOT.** To confirm trajectory results obtained using CellRank, we additionally used MOSCOT[72] (v0.3.3) to infer endocrine lineages using real time points. We ran MOSCOT on the porcine scRNA-seq data using time points E23, 33, and 40 (using geodesic distances, tau_a = 0.999, tau_b = 0.99999). We constructed an approximation of the geodesic distance by using an approximation of the Heat Kernel[122] and

constructed a KNN-graph with k = 15 based on a 50-dimensional PCA embedding. For time points E45, 63, 85 we ran MOSCOT on the multiome data with a joint embedding created by concatenating the scaled harmony embedding of the scRNA-seq data and scaled atac_poisson embedding. The inferred cell transitions are visualized using a Sankey diagram, excluding transitions that come from <5% of cells of one cell type.

**Human pluripotent stem cell culture and differentiation**
**hESC culture.** The NEUROG3[-/-] hESC line was previously created using CRISPR/Cas9 to disrupt endogenous expression with a frame-shift INDEL[55]. hESCs were maintained in mTeSR (StemCell Technologies) on hESC-qualified Matrigel (BD Biosciences) coated plates under standard culture conditions (37 °C, 5% $CO_2$, and 95% humidity). Cells were routinely passaged every four days with Dispase (Invitrogen). High-titer lentivirus with inducible NEUROG3 vectors was added to the media of newly plated hESCs. After 24 h, the media was replaced with mTeSR containing selective antibiotic G418 (500 mg/mL, Sigma). All transduced cell lines were maintained under selection.

**hESC differentiation.** hESCs were dispersed with Accutase (StemCell Technologies), washed, collected, resuspended in mTeSR containing 10 mM ROCK inhibitor (Y-27632, Tocris Bioscience), and plated at a concentration of $1 \times 10^5$ cells/cm² on Matrigel-coated, 24-well plates (Nunclon, Delta treated). When cells reached 75% confluency, differentiation was initiated. Day 0 medium was RPMI 1640 supplemented with non-essential amino acids, 100 ng/mL Activin A (Cell Guidance Systems) and 50 ng/mL BMP4 (R&D Systems). Days 1–2 medium contained 0.2% tetracycline-free FBS (Hyclone) but not BMP4. Days 3–4 medium was RPMI 1640 containing 2% FBS, 50 ng/mL FGF7 (R&D Systems), and 50 ng/mL Noggin (R&D Systems). Days 5–8 medium was high-glucose-DMEM (Gibco) containing 50 ng/mL Noggin, 2 mM all-trans retinoic acid (Stemgent), and 1% (0.5x) B27 without vitamin A (Gibco). Days 9–12 medium was high-glucose-DMEM supplemented with 1% B27 and 25 ng/mL Noggin.

**hiPSC culture.** hiPSCs were maintained in StemMACS™ iPS-Brew XF (iPS-Brew) culture medium (Miltenyi Biotec) on Geltrex (Gibco) coated dishes under standard culture conditions (37 °C, 5% $CO_2$ and 95% humidity). Cells were passaged every 4–5 days by single-cell dispersion using Accutase (Sigma-Aldrich). For aggregate suspension cultures, hiPSCs were detached with Accutase and seeded at a concentration of $0.8 \times 10^6$ cells/mL in iPS-Brew supplemented with Y-27632 to a 30-mL spinner flask (Reprocell) on a magnetic stirrer (Cultistir, Able) set at 60 rpm in a humidified 5% $CO_2$ 37 °C incubator. The aggregates were split every 3-4 days with Accutase.

**hiPSC differentiation.** To initiate differentiation, hiPSC aggregates were dispersed into single-cell suspension and seeded at $0.8 \times 10^6$ cells/ mL in a 30-mL spinner flask. Cells were cultured for 72 h in iPS-Brew and then differentiated towards pancreatic islets with a 6-stage protocol (detailed in Supplementary Data 10)[123]. Samples were collected on differentiation stage 5 day 7, stage 6 day 7 and stage 6 day 14.

**ChIP, RNA and ATAC-seq and data analysis**
**ChIP-seq.** Samples were collected on differentiation day 9 post 8 h NEUROG3 induction. Cells were cross-linked in 1% formaldehyde in PBS for 12 min at RT and were quenched by 0.125 M glycine. Nuclei were pelleted in lysis buffer (10 mM Tris-HCl pH 8.0, 10 mM NaCl, 0.2% NP-40). For chromatin fragmentation, cells were resuspended in Nuclear Lysis Buffer (20 mM Tris-HCl pH 8.0, 0.1% SDS, 2 mM EDTA) and sonicated in Diagenode Sonicator for 9 cycles of 20 s on, 60 s off at 4 °C. The desired amount of fragmented chromatin was supplemented with Nuclear Lysis Dilution Buffer (20 mM Tris-HCl pH 8.0, 0.1% SDS, 2 mM EDTA, 150 mM NaCl, 1% Triton X-100) and precleared with

blocked Protein G magnetic beads (Thermo Fisher Scientific) with rotation at 4 °C for 3 h. 1% precleared samples were saved as input and each 25 µg sample was incubated with 20 µg Protein G magnetic beads preloaded with 5 µg of NEUROG3 antibody (R&D) overnight with rotation at 4 °C. Beads were then washed sequentially at 4 °C using: (1) Washing A (150 mM NaCl, 20 mM Tris-HCl pH 8.0, 2 mM EDTA, 0.1% SDS, 1% Triton X-100, 0.1% sodium deoxycholate), (2) Serial Washing B (20 mM Tris-HCl pH 8.0, 2 mM EDTA, 0.1% SDS, 1% Triton X-100, 0.1% sodium deoxycholate) with 500 mM, 1 M and 2 M NaCl, (3) Washing C (50 mM Tris-HCl pH 8.0, 2 mM EDTA, 500Mm LiCl, 1% NP-40, 0.5% sodium deoxycholate) and final 2 times wash with TE (10 mM Tris-HCl pH 8.0, 10 mM EDTA). Protein-DNA complexes were eluted from the beads in Elution Buffer (50 mM Tris-HCl pH 8.0, 10 mM EDTA, 1% SDS) at 65 °C for 30 min. Cross-links were reversed at 65 °C overnight, and DNA was purified with phenol:chloroform (1:1), chloroform and ethanol for library construction and sequencing at CCHMC DNA sequencing and Genotyping Core Facility.

**ATAC-seq.** Samples were collected on differentiation day 9 and day 12 with or without 8 or 24 h NEUROG3 induction. Cells were dissociated with Accutase into a single-cell suspension. About 50,000 cells were collected for lysis and transposition based on the Omni-ATAC protocol. Briefly, cells were lysed with Lysis buffer (10 mM Tris-HCl pH 8.0, 10 mM NaCl, 3 mM $MgCl_2$, 0.1% NP-40, 0.1% Tween-20, 0.01% Digitonin), washed with washing buffer (10 mM Tris-HCl pH 8.0, 10 mM NaCl, 3 mM $MgCl_2$, 0.1% Tween-20), and incubated with Transposition buffer (Nextra) at 37 °C for 30 min rotated at 1,000 rpm. DNA was purified (Qiagen MinElute) for library construction and sequencing at the Kottyan lab, CCHMC.

**RNA Isolation and RNA-seq.** Samples were collected on differentiation days 9, 10, 11, and 12 with or without 8 h NEUROG3 induction. All RNA samples were column-purified using a NucleoSpin RNA kit (Macherey-Nagel) with an on-column DNase digestion according to the manufacturer's protocol. Frozen RNA samples were sent for library construction and sequencing at Novogene.

**NGS data processing.** ChIP-seq was performed at a depth of 30 M reads per sample. ATAC-seq was performed at a depth of 20 M reads per sample. RNA-seq was performed at 150 bp paired-end with a depth of 30 M reads per sample. Fastq read files for each sample were obtained and then aligned using the Computational Suite for Bioinformaticians and Biologists version 3.0 (CSBB-v3.0) to remove low quality bases and potential adapter contamination. Open chromatin region and transcription factor binding sites (NEUROG3) were then called, filtered, and annotated to the nearest gene using MACS2[124] (v2.1.0) and HOMER[125] (v4.11). Motif analysis annotations with genomic features and peak overlap were performed using HOMER, ChIPseeker[126] (v1.38.0) and ChIPpeakAnno[127] (v3.36.1). Raw transcript counts and normalized transcripts per million (TPM) values were obtained and analyzed for differential expression with DESeq2[128] (v1.42.1). For differential expression, statistical and biological significance was set at FDR < 0.05, log-fold-change > 1, with a minimum of 1 count from the triplicates of the total 12 samples (Supplementary Data 1). Directed GRN was constructed with the integration of ChIP, ATAC and RNA-seq data using mLASSO-STARS[129] based algorithm. Nodes of GRN were filtered with differentially expressed genes and protein function, whereas edges were pruned with interaction types and weights. Graph analysis on the constructed GRN was performed for cliques, coefficients, shortest paths, centrality, and community. The results were compared across sampling time points. Dynamic simulation and pseudo-interruption were conducted based on graph diffusion with the hyperbolic tangent as the activation function and as the filtration.

## Data sources
Published datasets are available publicly or accessible upon reasonable request from the corresponding authors of the original publications. Human data were from Yu et al.[41] (OMix (https://bigd.big.ac.cn/omix/) identifier OMIX236), Goncalves et al.[37] (European Genome-Phenome Archive (EGA, https://ega-archive.org/) ID# EGAD00001007506), de la O et al.[30] (database of Genotypes and Phenotypes (dbGaP) accession phs002693.v1.p1). Mouse data were from Bastidas-Ponce et al.[35] (Gene Expression Omnibus (GEO) accession GSE132188), Byrnes et al.[36] (GEO accession GSE101099), Han et al.[38] (GEO accession GSE136689), Krentz et al.[39] (GEO accession GSE120522), Yu et al.[42] (GEO accession GSE115931).

## Ethics
The use of pigs in this study was approved by the Committee on Animal Health and Care of the local government body of the state of Upper Bavaria in Germany for the wild-type and *INS*-eGFP[78] German Landrace pigs from the Ludwig Maximilian University of Munich (LMU, Permission No. 55.2-2532.Vet_02-17-136) and the *PTF1A*-codon-improved Cre (iCre)[79] and *ROSA*-mTmG[80] pigs from the Technical University of Munich (TUM, Permission No. 55.2-2532.Vet_02-18-33). Experiments were conducted in accordance with the German Animal Welfare Act and Directive 2010/63/EU on the protection of animals used for scientific purposes.

The parent human embryonic stem cell (hESC) line WA01 (H1) was obtained from WiCell. All experiments with hESCs were approved by the Cincinnati Children's Hospital ESCRO committee (Protocol# EIPDB2713).

The human-induced pluripotent stem cell (hiPSC) line HMGUi001-A[81] was generated at Helmholtz Munich. All experiments with hiPSCs were approved by the Ethics Committee of the Technical University Munich (219/20 S, 290/20 S).

## Reporting summary
Further information on research design is available in the Nature Portfolio Reporting Summary linked to this article.

## Data availability
The data generated in this study have been deposited in NCBI's Gene Expression Omnibus. The pig pancreas data is accessible through GEO Series accession number GSE262280. The hESC datasets are accessible through GSE261950, GSE261951, and GSE261952. Sequencing data from pig pancreas were aligned using the Sscrofa11.1 assembly of the pig genome (https://www.ebi.ac.uk/ena/browser/view/GCA_000003025.6) and the improved annotation based on the Ensembl annotation version 101 (see Methods; Improved pig gene annotation file is available at: https://github.com/theislab/pig-embryo-ana). Sequencing data from hESC were aligned using the GRCh37/hg19 reference genome and Ensembl gene annotation (https://www.ncbi.nlm.nih.gov/datasets/genome/GCF_000001405.13/). Any other data supporting the findings of this study are available from the corresponding authors on reasonable request.

## Code availability
Jupyter notebooks to reproduce the analysis and figures are available at: https://github.com/theislab/pig-embryo-ana[130].

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

## Acknowledgements

The authors thank S. Schirge, A. Bastidas-Ponce, L. Appel, I. Kunze, K. Diemer (IDR, Helmholtz Munich), C. Blechinger, M. Krätzl (CiMM, Ludwig Maximilian University of Munich) and D. Kalla (Technical University of Munich) for their excellent technical assistance; D. Kechele (Cincinnati Children's Hospital Medical Center) for great support in organizing and depositing next generation sequence data; M. Bakhti and A. Böttcher (IDR, Helmholtz Munich) for helpful discussions; A. Grapin-Botton, J. Stratmann and C.A. Gonçalves (Max Planck Institute of Molecular - Cell Biology and Genetics) for generously sharing human fetal pancreas data; D. Klein, M. Lange, V. Bergen, and A. Moinfar (ICB, Helmholtz Munich) for bioinformatic analysis advice; the animal caretakers in the pig facilities in Ludwig Maximilian University of Munich and Technical University of Munich. The authors of this study received funding from the European Research Council (grant 101054564 – to H.L. and grant 101054957 – to F.J.T), the German Research Foundation (LI 1006/3-1– to H.L., TH 900/13-1 – to F.J.T, WO 685/22-1 – to E.W., Transregio Research Unit 127 and 205 funding – to E.K. and E.W., Collaborative Research Centre 1321 funding to A.S. and T.F.), the Juvenile Diabetes Research Foundation (grant 3-SRA-2023-1420-S-B – to H.L, E.W., F.J.T, grant 2-SRA-2019-773-S-B – to J.B.S.), the German Federal Ministry of Education and Research to the German Centre for Diabetes Research (grant 82DZD00802 – to E.W.), the National Institutes of Health (grant R01DK118421 – to J.B.S, grant R25GM056847-23 – to S.O., grant R01DK118421-02S1 – to S.O.), the Kraft Family Fellowship to the UCSF Diabetes Center (S.O.).

## Author contributions

K.Y. and H.L. designed the study. H.S. and F.J.T. designed bioinformatic analysis plans. K.Y., H.S., and H.L. wrote the manuscript. B.K., E.W., and E.K. advised on the study design and provided pig pancreas samples. K.Y. and M.S. prepared samples for single-cell genomics. K.Y., H.S., M.S., and K.H. performed bioinformatic analyses. S.O. and J.B.S provided human fetal data and analysis advice. X.Z. designed and performed hESC differentiation experiments and analyzed the data, supervised by J.M.W., K.Y., E.S., and K.S. designed and performed hiPSC differentiation experiments and analyzed the data. M.U. and T.W. improved pig genome annotations and sequenced the libraries. K.F. and T.F. provided pig pancreas samples, supervised by A.S. H.L. provided financial support. F.J.T. and H.L. supervised the study. All authors reviewed and edited the manuscript.

## Funding

## Competing interests

F.J.T. consults for Immunai Inc., Singularity Bio B.V., CytoReason Ltd, Cellarity, Curie Bio Operations, LLC, and has an ownership interest in Dermagnostix GmbH and Cellarity. The remaining authors declare no competing interests.

## Additional information

[1]Institute of Diabetes and Regeneration Research (IDR), Helmholtz Munich, Neuherberg, Germany. [2]German Center for Diabetes Research (DZD), Neuherberg, Germany. [3]Institute of Computational Biology (ICB), Helmholtz Munich, Neuherberg, Germany. [4]Institute for Stroke and Dementia Research, University Hospital, Ludwig Maximilian University of Munich, Munich, Germany. [5]Department of Mathematics, Technical University of Munich, Munich, Germany. [6]Department of Cell and Tissue Biology, University of California, San Francisco, USA. [7]Diabetes Center, University of California, San Francisco, USA. [8]Eli and Edythe Broad Center of Regeneration Medicine and Stem Cell Research, University of California, San Francisco, USA. [9]Division of Developmental Biology, Cincinnati Children's Hospital Medical Center, Cincinnati, USA. [10]Center for Stem Cell and Organoid Medicine (CuSTOM), Cincinnati Children's Hospital Medical Center, Cincinnati, USA. [11]Core Facility Genomics, Helmholtz Munich, Neuherberg, Germany. [12]Chair of Livestock Biotechnology, Department of Molecular Life Sciences, School of Life Sciences, Technical University of Munich, Freising, Germany. [13]Division of Endocrinology, Cincinnati Children's Hospital Medical Center, Cincinnati, USA. [14]Chair for Molecular Animal Breeding and Biotechnology Gene Center, Ludwig Maximilian University of Munich, Munich, Germany. [15]Center for Innovative Medical Models (CiMM), Ludwig Maximilian University of Munich, Munich, Germany. [16]School of Life Sciences Weihenstephan, Technical University of Munich, Freising, Germany. [17]School of Medicine, Technical University of Munich, Munich, Germany. [18]These authors contributed equally: Kaiyuan Yang, Hannah Spitzer. ✉e-mail: fabian.theis@helmholtz-munich.de; heiko.lickert@helmholtz-munich.de

