## [Peer Review file · Nature Communications]

A multimodal cross-species comparison of pancreas development

Corresponding Author: Professor Heiko Lickert

Version 0:

Reviewer comments:

Reviewer #1

(Remarks to the Author)

In the revised manuscript, the authors have comprehensively addressed all of my concerns. By including extended computational analyses and in situ validation data, more detailed information on evolutionary conservation of pancreas development across pig, human, and mouse is provided, and the comparison is reported in a more balanced manner. This revised study represents an important and useful resource introducing pig as a powerful alternative model for pancreas development with advantages over alternative models such as mouse or human in vitro systems.

Reviewer #2

(Remarks to the Author)

I would like to thank the authors for their comprehensive and point-by-point responses to all of the comments raised in the first review. The revised manuscript demonstrates clear improvements in the following areas: (1) avoiding the assertion that the pig is “superior” to the mouse, instead positioning it as a complementary platform; (2) providing detailed genotype and enrichment strategies for the transgenic pig; (3) expanding the analysis of human and pig gene-regulatory networks (GRNs); and (4) including additional experimental validation for the PEC population. Furthermore, all minor comments have been effectively addressed. However, several substantive issues remain that require attention and must be addressed before publication.

1. The exact transgenic pig lines used in each figure should still be explicitly stated, either in the figure legends or the main text. This detail is critical for reproducibility and clarity.
2. The metrics for doublet detection should be displayed in the manuscript, not just described in the Methods section. Additionally, the distribution of detected gene counts and read counts across clusters needs to be reported in order to rule out potential technical artifacts.
3. The developmental origin of PEC as a bona fide first-wave endocrine progenitor remains unresolved. Trajectory analysis beyond UMAP/CellRank is needed to substantiate this claim. Furthermore, enrichment scores for the proposed non-housekeeping PEC markers (e.g., ASPM, ECT2) in relation to other pancreatic cell types are missing. Figure 5f/g shows sparse ASPM expression in PEC and notable levels in MPC; this overlap should be quantified. Additionally, the sysVI integration splits pig PP cells into two subclusters, with divergent placements in the pig-human versus pig-mouse spaces. The authors should clarify whether this split reflects biological heterogeneity or technical noise. Moreover, they need to demonstrate whether the EP-beta/PEC connection observed in the pig-mouse UMAPs is reproducible in mouse-only data. Finally, the proposed PEC-like cells in both human and mouse remain transcriptome-level constructs, and functional validation, such as β -cell differentiation capacity, is still lacking.
4. The failure to resolve PP and alpha clusters in the mouse atlas (Figure 2a) contradicts findings from published datasets (e.g., Ref. 42) and warrants further explanation. Additionally, mouse epsilon cells were not included in the CellRank trajectories (Figure 3a). Both mouse developmental GRNs and NEUROG3 networks were also not presented alongside the corresponding data for pig and human (Figures 2f, 3g,h), which should be addressed.
5. While the authors now present the pig as a complementary model, they have not specified which regulatory mechanisms or cellular states are uniquely illuminated by pig-human integration, as opposed to being obscured in the pig-mouse

comparison. This key distinction needs to be made clearer. This is also central to the focus of the article.

6. The reliance on in-vitro hESC NEUROG3 CUT&RUN data to validate in-vivo pig and human GRNs remains problematic due to known systemic differences. In-vivo NEUROG3 binding data (via CUT&RUN or CHIP-seq) or quantitative overlap statistics are still needed to substantiate the claimed conservation between species.

7. This comment has been satisfactorily addressed.

8. This comment has been satisfactorily addressed.

9. The β_0/β_1 distinction in β -cells—does this reflect the distinction between the first wave and second wave of endocrine cells?

10. The Louvain clustering driven by MEIS2 expression lacks demonstrated biological significance or cross-species conservation. In particular, β_0 cells do not integrate well between pig and human datasets. Additionally, the extent and temporal distribution of human β -cell heterogeneity (Extended Data Figure 7d/f) diverges markedly from the pig pattern (Extended Data Figure 7b).

Minor Comment:

Figure 2e annotates a human cell cluster as “PEC,” which needs to be corrected.

Reviewer #3

(Remarks to the Author)

In this revised version, the authors have significantly extended the in silico investigations and performed additional experiments, including immunostainings for NR2F2 to better characterize the new PEC population. The authors have satisfactorily addressed most of my comments. They notably clarified the expression of Meis2 over time and its relation to beta 0 and beta 1 populations. Though the meaning of these two populations with regards to function and relation to other populations remains to be clarified this is beyond the scope of this already extensive study.

One aspect that remains unclear is the PEC population. The authors clarified that it was present in the E23_2 sample comprising 4 embryos (70 cells) and the E22 sample comprising 6 embryos (4 cells). They eliminated E23_1 from the study as it had too few pancreatic cells. From the current text, it took me a while to understand that the PEC population was not restricted to E22 and 23 but also persisted later (the resolution of Figure 1i on the reviewer pdf is too low to distinguish rare populations and this figure panel is very small). The sentence line 207 could be made clearer, specifying that these cells were detected at all stages. While it is difficult to understand what this population is (it does not seem to stem from any progenitor cluster) and there may be a technical reason behind this population (or a stage in proliferation- the expression of histone 2AZ is puzzling) the authors should be free to highlight it as they did. The authors also identified NR2F2 as an additional marker for the early PECs. It should be made clear that this marker is not specific as it also marks early progenitors more broadly as well as mesenchymal cells (notably line 648).

Detailed comments:

Line 197: It may not be judicious to call the cluster NGN3. First, because none of the other clusters have been called by a gene name. Second, because these cells have traditionally been called EP for endocrine progenitors. It is actually referred to as EP by the authors in mouse and human in Figure 2a.

Line 345: “... a large group of...”?

Author Responses to Comments:

Summary

We sincerely thank the reviewers for their positive assessment of our revised manuscript and their valuable feedback. Their insightful comments have been instrumental in enhancing the clarity and quality of our work. Our point-by-point responses are provided in blue text. Where applicable, we have cited revised manuscript text *in italics*, provided new or revised figures, and indicated the locations of all changes. Line numbers correspond to the clean revised manuscript file. Please note that reference numbers cited in this rebuttal may have changed in the final manuscript.

REVIEWERS' COMMENTS

Reviewer #1 (Remarks to the Author):

In the revised manuscript, the authors have comprehensively addressed all of my concerns. By including extended computational analyses and in situ validation data, more detailed information on evolutionary conservation of pancreas development across pig, human, and mouse is provided, and the comparison is reported in a more balanced manner.

This revised study represents an important and useful resource introducing pig as a powerful alternative model for pancreas development with advantages over alternative models such as mouse or human in vitro systems.

We thank the reviewer for the positive assessment of our revised manuscript and for the generous comments. It is rewarding to know that the reviewer considers our study a balanced and important resource for the field.

Reviewer #2 (Remarks to the Author):

I would like to thank the authors for their comprehensive and point-by-point responses to all of the comments raised in the first review. The revised manuscript demonstrates clear improvements in the following areas: (1) avoiding the assertion that the pig is “superior” to the mouse, instead positioning it as a complementary platform; (2) providing detailed genotype and enrichment strategies for the transgenic pig; (3) expanding the analysis of human and pig gene-regulatory networks (GRNs); and (4) including additional experimental validation for the PEC population. Furthermore, all minor comments have been effectively addressed. However, several substantive issues remain that require attention and must be addressed before publication.

We are encouraged by the reviewer's acknowledgment that the manuscript has been clearly improved in the highlighted key areas. We have carefully considered all remaining substantive issues raised and have prepared a comprehensive response below, detailing the additional revisions we have made to the manuscript to address each concern.

1. The exact transgenic pig lines used in each figure should still be explicitly stated, either in the figure legends or the main text. This detail is critical for reproducibility and clarity.

We agree with the reviewer that specifying the pig lines used is critical for reproducibility and clarity. Accordingly, we have stated the specific pig line information for each experiment in the legends of all relevant figures.

As an example, the added text in the legend for Figure 2 now states: "*a-c: scRNA-seq of pancreatic cells from wild-type and INS-eGFP¹ pigs. d-f: Multiome analysis of pancreatic cells from PTF1A-codon-improved-Cre/ROSA-mTmG pigs. Detailed sample information is provided in Supplementary Data 1.*"

2. The metrics for doublet detection should be displayed in the manuscript, not just described in the Methods section. Additionally, the distribution of detected gene counts and read counts across clusters needs to be reported in order to rule out potential technical artifacts.

We thank the reviewer for highlighting the need for these important quality metrics. We have included the doublet detection metrics in the main text (Lines 129-131) and supplied the distribution of detected gene and read counts across clusters as a new Supplementary Data 13 and in Supplementary Figure 2e,f. We believe these additions provide full transparency into the quality of our data and effectively rule out potential technical artifacts.

The updated text in the manuscript (Lines 129-131) and figure (Supplementary Fig. 2e,f) are as follows:

Lines 129-131: "*The resulting dataset was subjected to stringent doublet removal: cells consistently identified as doublets by >3 among the six methods used were removed, and entire clusters with a doublet frequency >70% were excluded (see Methods for details)*"

Supplementary Fig. 2e,f. Violin plots showing distribution of the detected read counts (e) and gene counts (f) across clusters (Supplementary Data 13).

3. The developmental origin of PEC as a bona fide first-wave endocrine progenitor remains unresolved. Trajectory analysis beyond UMAP/CellRank is needed to substantiate this claim. Furthermore, enrichment scores for the proposed non-housekeeping PEC markers (e.g., ASPM, ECT2) in relation to other pancreatic cell types are missing. Figure 5f/g shows sparse ASPM expression in PEC and notable levels in MPC; this overlap should be quantified. Additionally, the sysVI integration splits pig PP cells into two subclusters, with divergent placements in the pig-human versus pig-mouse spaces. The authors should clarify whether this split reflects biological heterogeneity or technical noise. Moreover, they need to demonstrate whether the EP-beta/PEC connection observed in the pig-mouse UMAPs is reproducible in mouse-only data. Finally, the proposed PEC-like cells in both human and mouse remain transcriptome-level constructs, and functional validation, such as β -cell differentiation capacity, is still lacking.

We thank the reviewer for these insightful and constructive comments. We have addressed each point as detailed below.

1) Trajectory analysis:

We agree with the reviewer on the importance of using multiple methods to verify trajectory inferences. In addition to the pseudo-temporal approach of CellRank, we applied the MOSCOT framework (Lines 379-385, Figure 6e) to reconstruct trajectories based on real developmental time points. To map the entire developmental timeline comprehensively, we used MOSCOT on both our multiome data (E45-E85) to identify the trajectory from PECs to beta cells from E45 onward, and on our scRNA-seq dataset to successfully reconstruct the early trajectory from E20. This application of an independent method provided additional confirmation of the developmental trajectories from PECs to beta cells. We have added Figure 6g to highlight this key confirmation from multiple methods.

2) PEC marker analysis (ASPM/ECT2):

We thank the reviewer for this suggestion. We have now provided the differential expression analysis results comparing PEC cluster to all other pancreatic cell types as a new Supplementary Data 14. These two genes appear as significantly differentially expressed genes (adjusted P-value < 0.05, \log_2 fold change >1) not only in the comparison between PEC and MPC but also in comparisons to all other clusters.

3) sysVI PP cell division:

The reviewer is correct to note the interesting split within the pig PP cell population. We have carefully examined the marker gene expression. However, our analysis did not reveal clear marker genes to robustly distinguish these two sub-populations of pig PP cells. Instead, we found that (as shown below): 1) pig PP cells have marginal expression of certain human epsilon marker genes; 2) certain genes enriched in pig

PP cells have higher expression in human epsilon cells than in human PP cells. Therefore, we conclude that this combination of factors is the most likely driver for the algorithm to align a sub-population within the pig PP cells with the human epsilon cells.

Marker gene expression in pig (sysVI-embedding)

Marker gene expression in human (sysVI-embedding)

4) EP-beta/PEC connection in mouse-only data:

We have mapped the mouse PEC-like cells (identified via sysVI) onto our mouse embryonic pancreas atlas (as shown below). We observe that these mouse PEC-like

cells are located at the intersection of endocrine progenitors and beta cells, suggesting a possible EP-beta/PEC connection within the mouse developmental landscape.

5) Functional validation:

We fully agree with the reviewer that functional validation of the β -cell differentiation capacity of PECs is a critical next step. As our current study is primarily focused on establishing a comprehensive multi-species transcriptional atlas and identifying evolutionarily conserved progenitor populations *in vivo*, the derivation of primary PECs and their functional differentiation assays is a key objective for our future work. We have acknowledged this limitation in the revised Discussion (Lines 427-428): “*However, definitive validation of this pathway and the differentiation capacity of PECs remains a crucial goal for future work.*”

4. The failure to resolve PP and alpha clusters in the mouse atlas (Figure 2a) contradicts findings from published datasets (e.g., Ref. 42) and warrants further explanation. Additionally, mouse epsilon cells were not included in the CellRank trajectories (Figure 3a). Both mouse developmental GRNs and NEUROG3 networks were also not presented alongside the corresponding data for pig and human (Figures 2f, 3g,h), which should be addressed.

We thank the reviewer for these insightful observations, which allow us to clarify several important points regarding the mouse data and our analytical choices.

1) Resolution of PP and alpha cells in the mouse atlas:

We agree with the reviewer that PP and alpha cells are distinct cell types. The lack of obvious separation between these clusters in our presented integration is due to two technical factors:

- a) Low abundance of PP cells: The available public datasets we integrated had very few PPY-expressing cells in comparison to other endocrine cells. This limits the power to resolve them as a fully independent cluster in the final UMAP.
- b) Shared gene expression programs: PP and alpha cells share a panel of key genes (e.g., *Cpe*, *Isl1*, *Arx*), which likely also contributes to their close localization in the integrated latent space. As shown in Supplementary Figure 4a and below, a very small, putative PP cell cluster (expressing *Scg2* and *Ppy* and circled in dark) is located adjacent to the dominant alpha cell cluster.

2) Inclusion of epsilon cells in trajectories:

The reviewer is correct that epsilon cells were not highlighted in the main Figure 3a due to the following two reasons:

- a) Our Figure 2 specifically focuses on the beta and alpha endocrine lineages, which are the most prominent and well-characterized in our pig atlas and are the central theme of that results section.
- b) Epsilon cells were not captured in our pig dataset (as shown below), therefore we didn't perform cross-species comparison of the epsilon lineage.

The CellRank analysis on the entire integrated human and mouse atlases including epsilon cells is provided in Supplementary Figure 6a,b.

3) Presentation of mouse GRNs and NEUROG3 networks:

We thank the reviewer for this suggestion. The generation of the mouse multiome data (scRNA-seq + scATAC-seq) necessary for reconstructing developmental GRNs and the NEUROG3 regulon is indeed critical. This work was conducted as part of a separate, extensive study focused specifically on cross-species gene regulatory networks. The corresponding manuscript, which provides a comprehensive comparison of these GRNs between mouse and human, is currently in its final stages and nearing submission. We believe it will serve as a perfect complement to the resource presented here.

5. While the authors now present the pig as a complementary model, they have not specified which regulatory mechanisms or cellular states are uniquely illuminated by pig-human integration, as opposed to being obscured in the pig-mouse comparison. This key distinction needs to be made clearer. This is also central to the focus of the article.

We thank the reviewer for this critical comment, which allows us to better highlight the central findings of our work. As suggested, we have updated the text in the Discussion (Lines 410-429) to state the unique pig-human conserved regulatory mechanisms and cellular states, while carefully framing these findings in the context of pig as a complementary model for studying human development.

The updated text in the Discussion (Lines 410-429) is as follows:

Lines 410-429: “... *First, the resemblance in developmental tempo of pig and human gestation provides a temporally aligned framework to study extended pancreas organogenesis events that are compressed in the mouse. Second, we observe a pig-human conservation in epigenetic and transcriptional regulation, particularly in the endocrine lineage. The high conservation of transcription factors downstream of NEUROG3 (over 50% between pig and human) suggests a core program for endocrine fate acquisition in larger mammals. This aligns with both the recent human study² and the observed similarities in postnatal islet characteristics, fasting C-peptide and glucose levels between pigs and humans^{3,4}. By leveraging the temporally resolved single-cell multiomic pig pancreas atlas covering all three trimesters, we identified a unique primed endocrine cell (PEC) population, potentially representing a progenitor state that is distinct from the classic NGN3 endocrine progenitor cluster. Transcriptionally matched PEC-like cells were also identified in human and mouse, with murine PEC-like cells showing a possible link to the NGN3 cluster. Concurrent with pancreas morphogenesis, we discovered that both NGN3 and PEC clusters emerged as heterogeneous populations and were predicted to hold dynamic lineage potential over time. This conserved PEC population is intriguing as it may suggest a potential NEUROG3-independent pathway for endocrinogenesis, which could offer an explanation for the persistence of endocrine cells in some human patients carrying homozygous NEUROG3 mutations^{5,6}. However, definitive validation of this pathway and the functional capacity of PECs remains a crucial goal for future work. We further identified heterogeneous beta cells, suggesting the acquisition of islet cell heterogeneity may originate in embryonic development. ...”*

6. The reliance on in-vitro hESC NEUROG3 CUT&RUN data to validate in-vivo pig and human GRNs remains problematic due to known systemic differences. In-vivo NEUROG3 binding data (via CUT&RUN or ChIP-seq) or quantitative overlap statistics are still needed to substantiate the claimed conservation between species.

We thank the reviewer for raising this crucial point regarding the limitations of *in vitro* systems, which rightly highlights a fundamental challenge in the field. Our use of the in vitro hESC and hiPSC data as the best currently available proxy was intended to provide insight into conserved regulatory mechanisms across model systems.

Furthermore, we fully agree that *in vivo* binding data is crucial. To directly overcome this limitation and generate the definitive *in vivo* data, we have established a collaboration with colleagues in France where human embryonic samples are accessible. This project is currently underway, and we anticipate that it will provide the key validation suggested by the reviewer in future work.

7. This comment has been satisfactorily addressed.

We thank the reviewer for this positive feedback.

8. This comment has been satisfactorily addressed.

We thank the reviewer for this positive feedback.

9. The β_0/β_1 distinction in β -cells—does this reflect the distinction between the first wave and second wave of endocrine cells?

We thank the reviewer for this insightful question. We agree with the reviewer that our data suggests a connection between these clusters and the waves of endocrinogenesis. This temporal pattern indeed suggests that the β_0 cluster contains the early-pioneer endocrine cells originating from the first wave, which may persist throughout development, though we cannot rule out that some β_0 cells are also produced later. In contrast, the β_1 cluster is predominantly formed during the secondary transition and likely represents the major population of maturing, expanding beta cells. Therefore, we have described the temporal pattern of the two clusters as follows: (Lines 298-299) *'The Beta,0 cluster consisted of cells appearing at all stages, whereas the Beta,1 cluster emerged mainly during the second wave of endocrinogenesis.'*

10. The Louvain clustering driven by MEIS2 expression lacks demonstrated biological significance or cross-species conservation. In particular, β_0 cells do not integrate well between pig and human datasets. Additionally, the extent and temporal distribution of human β -cell heterogeneity (Extended Data Figure 7d/f) diverges markedly from the pig pattern (Extended Data Figure 7b).

We appreciate the reviewer's critical comment, which provides an opportunity to clarify our analysis and address these concerns.

1) Regarding the Louvain clustering:

We agree that the Louvain algorithm identifies clusters based on transcriptional similarity and that the biological role of MEIS2 requires further validation. Our goal was to test if unbiased computational algorithms could identify subtypes within the human beta cells. The resulting clusters were associated with MEIS2 expression. We have ensured that the text presents MEIS2 as a potential marker for beta cell heterogeneity in humans; its function will be a subject of our follow-up studies.

2) Regarding pig-human integration of β_0 cells:

The reviewer is correct to note the integration challenges. This is likely due to species-specific differences in their global transcriptional programs, as suggested by a cluster correlation score of 0.64 (Figure 2b) based on ortholog gene expression profiles between pig and human beta cells. Despite this divergence, in our differential expression analysis comparing beta subtypes (stratified by *MEIS2* expression), we observed conserved gene signatures and pathways, suggesting the features defining beta cell subtypes are shared between pig and human.

3) Regarding the temporal distribution of β cell heterogeneity:

We agree that the temporal pattern of β cell heterogeneity appears to diverge, and we believe this reflects a fundamental biological difference in developmental timing. As illustrated in Fig. 1a, the major wave of β cell differentiation in humans occurs after the developmental equivalent of pig E40 (the "secondary transition" in pig). Therefore, we would expect the temporal pattern of β cell heterogeneity in humans to mirror the pattern observed in the pig during the E40-85 window. The human pattern in Fig. 5d/f is consistent with this expected timeline.

Minor Comment:

Figure 2e annotates a human cell cluster as "PEC," which needs to be corrected.

We thank the reviewer for their careful reading. The annotation "PEC (ss)" indicates this cluster is from the pig (*Sus scrofa*) data, as detailed in the figure legend. The human clusters are correspondingly labeled with an "hs" in the brackets. To prevent any potential confusion, we have ensured this species prefix notation is stated in the main figure panel.

Reviewer #3 (Remarks to the Author):

In this revised version, the authors have significantly extended the in silico investigations and performed additional experiments, including immunostainings for NR2F2 to better characterize the new PEC population. The authors have satisfactorily addressed most of my comments. They notably clarified the expression of Meis2 over time and its relation to beta 0 and beta 1 populations. Though the meaning of these two populations with regards

to function and relation to other populations remains to be clarified this is beyond the scope of this already extensive study.

We thank the reviewer for the positive feedback on our extensive revisions, including the new immunostainings for NR2F2 and the extended *in silico* analyses. We are glad that the clarifications regarding Meis2 expression were found to be satisfactory. We also agree that defining the precise function and interrelation of the beta 0 and beta 1 populations is a critical biological question that lies beyond the scope of this current work but represents an exciting avenue for future investigation.

One aspect that remains unclear is the PEC population. The authors clarified that it was present in the E23_2 sample comprising 4 embryos (70 cells) and the E22 sample comprising 6 embryos (4 cells). They eliminated E23_1 from the study as it had too few pancreatic cells. From the current text, it took me a while to understand that the PEC population was not restricted to E22 and 23 but also persisted later (the resolution of Figure 1i on the reviewer pdf is too low to distinguish rare populations and this figure panel is very small). The sentence line 207 could be made clearer, specifying that these cells were detected at all stages. While it is difficult to understand what this population is (it does not seem to stem from any progenitor cluster) and there may be a technical reason behind this population (or a stage in proliferation- the expression of histone 2AZ is puzzling) the authors should be free to highlight it as they did. The authors also identified NR2F2 as an additional marker for the early PECs. It should be made clear that this marker is not specific as it also marks early progenitors more broadly as well as mesenchymal cells (notably line 648).

We thank the reviewer for this helpful comment and for pointing out these areas where we can improve the clarity and precision of our description.

Regarding the persistence of the PEC population, we apologize for the lack of clarity on this point. We have now revised the text to explicitly state that rare PEC cells were detected at all developmental stages analyzed, not only at E22 and E23 (Lines 146-149). We have also ensured that the PEC population is more clearly labeled and visible in a higher-resolution version of Figure 1i for the final publication.

Regarding the specificity of NR2F2, we agree completely with the reviewer that NR2F2 is not a unique marker for PECs and is also expressed in early progenitors and mesenchymal cells. Our intention was to present it as an additional marker that can be used to identify this population in combination with other markers, such as E-Cadherin. To avoid any misunderstanding, we have revised the text to clarify this point and to reference its broader expression pattern (Lines 372-373).

The updated text in the manuscript is as follows:

Lines 146-149: *“A primed endocrine cell (PEC) cluster first emerged at E23 alongside sparse NEUROG3⁺ endocrine progenitors. This population, which persisted throughout all subsequent*

stages, exhibited features of endocrine cells and expressed genes coding for cytoskeletal components (TUBA1B, TUBB, STMN1) and cell cycle regulators (H2AFZ, PCLAF)."

Lines 372-373: *"Notably, we identified NR2F2 as a key transcription factor highly expressed in the 1°PEC cluster, though it was also detected in broader early progenitors and mesenchymal cells (Extended Data Fig. 8c)."*

Detailed comments:

Line 197: It may not be judicious to call the cluster NGN3. First, because none of the other clusters have been called by a gene name. Second, because these cells have traditionally been called EP for endocrine progenitors. It is actually referred to as EP by the authors in mouse and human in Figure 2a.

We thank the reviewer for raising this point, which allows us to clarify a key aspect of our analysis. The reviewer is correct that we use the term 'EP' for the human cluster.

In both our pig and mouse datasets (Supplementary Fig. 4f), we identified two distinct populations: 1) a classic *NEUROG3*-high endocrine progenitor (EP) cluster, and 2) a *FEV*-high-*NEUROG3*-low endocrine precursor cluster. To accurately represent this biological difference, we named the clusters after their most defining marker ('NGN3' for the progenitors, 'FEV' for the precursors).

As noted by the reviewer, the human data differs. *FEV*-expressing cells maintained high *NEUROG3* expression and formed a single, resolvable population (Supplementary Fig. 4b), which we therefore annotated broadly as 'EP'. We have included the explanation for the difference on Lines 165-166: "*FEV*⁺ cells in human were annotated together with the NGN3 cluster as EP (endocrine progenitor) cluster due to high *NEUROG3* expression (Supplementary Fig. 4b)." To ensure this important distinction is clear, we have added more details when describing pig *FEV* cluster on Lines 135-136: "*FEV* (*FEV/CHGB*, endocrine precursors with low *NEUROG3* expression)"

Line 345: "... a large group of...?"

We appreciate the reviewer's attention to detail. The text has been updated to: "*a large group of endocrine TF modules was conserved between pig and human, ...*" (Lines 229-230).

References

- 1 Kemter, E. *et al.* INS-eGFP transgenic pigs: a novel reporter system for studying maturation, growth and vascularisation of neonatal islet-like cell clusters. *Diabetologia* **60**, 1152-1156 (2017). <https://doi.org:10.1007/s00125-017-4250-2>
- 2 de la O, S. *et al.* Single-Cell Multi-Omic Roadmap of Human Fetal Pancreatic Development. *bioRxiv* (2022). <https://doi.org:10.1101/2022.02.17.480942>
- 3 Kim, S. *et al.* Molecular and genetic regulation of pig pancreatic islet cell development. *Development* **147** (2020). <https://doi.org:10.1242/dev.186213>
- 4 Renner, S. *et al.* Porcine models for studying complications and organ crosstalk in diabetes mellitus. *Cell Tissue Res* **380**, 341-378 (2020). <https://doi.org:10.1007/s00441-019-03158-9>
- 5 Pauerstein, P. T. *et al.* Dissecting Human Gene Functions Regulating Islet Development With Targeted Gene Transduction. *Diabetes* **64**, 3037-3049 (2015). <https://doi.org:10.2337/db15-0042>
- 6 Rubio-Cabezas, O. *et al.* Neurogenin 3 is important but not essential for pancreatic islet development in humans. *Diabetologia* **57**, 2421-2424 (2014). <https://doi.org:10.1007/s00125-014-3349-y>
- 7 Schreiber, V. *et al.* Extensive NEUROG3 occupancy in the human pancreatic endocrine gene regulatory network. *Mol Metab* **53**, 101313 (2021). <https://doi.org:10.1016/j.molmet.2021.101313>